# FLIPS: Instance-Fingerprinting for LLMs via Pseudo-random Sequences

**Gurvan Richardeau** [1]  **Gohar Dashyan** [1]  **Erwan Le Merrer** [2]  **Gilles Tredan** [3]

## Abstract

Literature reveals that a Large Language Model's (LLM) behavior is not only conditioned by its original weights but also its instance-level parameters, such as instructional prompt, sampling configuration or quantization. A model that generates safe outputs under one configuration may produce toxic content under another. However, current LLM identification techniques (such as fingerprinting) focus on intellectual property protection, and their design favors robustness to changes in these instance-level parameters. This poses a critical challenge for AI regulation in which compliance assessments target actual deployed behaviors, not model provenance. In this paper, we introduce instance-level fingerprinting, a regulator-oriented paradigm that distinguishes configurations of the same LLM. Our method, FLIPS, exploits biases in generated binary random sequences to reach 96% (closed-set) and 90% (open-set, where some targets are unknown) identification accuracy across 237 model instances, versus 35% for the adapted LLMmap baseline. This shows that instance-level fingerprinting is both necessary for regulation and practically feasible. Code available at [1].

## 1. Introduction

Identifying Large Language Models (LLMs) in a black-box setting has recently gained significant traction. LLM Fingerprinting aims to determine whether a *reference model* and a *target model* are identical. It consists of extracting a signature (fingerprint) from the reference model (*extraction*) and comparing it with one from the target model (*verification*) (Shao et al., 2025).

But what defines identical models? Prior work (Pasquini et al., 2025; Jin et al., 2024; Yang & Wu, 2024; Gubri et al., 2024; Zhu et al., 2025) focuses on Intellectual Property Protection (IPP) scenarios in which the target model is suspected to be a (possibly unauthorized) copy or derivative of the reference model. In that setting, a *model* is typically defined by the original weights i.e., resulting from a training procedure (since this constitutes the primary cost), and close derivatives (e.g., fine-tuning, system prompts, quantizations) are *not* considered as distinct models. In this work, we shift from this IPP scenario to consider regulatory scenarios in which the compliance of an LLM's behavior is evaluated. This is motivated by growing legislation surrounding AI governance, most notably the EU AI Act[2], and more recently California's new AI transparency law, SB 53[3].

This requires sensitivity to factors beyond original weights, which the literature indicates to be instance-level parameters such as system prompts, sampling configurations, quantization or fine-tuning. We then introduce *Instance-level Fingerprinting* (IF) that targets instances defined as a model along with its precise weights and configuration. Such a fingerprinting method allows regulators to efficiently identify known instances served by any provider, avoiding costly full audits. Critically, IPP-oriented methods deliberately ignore instance-level parameters to achieve robustness, rendering them blind to behavioral modifications. As shown in Figure 1, the state-of-the-art IPP method LLMmap conflates abliterated[4] models with their original counterparts. In contrast, our method, FLIPS, explicitly designed to be sensitive to such changes, avoids this confusion.

**Parameters Affecting LLM's Behavior**   Literature shows that system prompts can alter model behavior, Neumann et al. (2025) shows that system prompts such as "you are a Christian" lead to very different definitions of heterosexuality, while Mou et al. (2024); Cui et al. (2024) show that a safe-oriented system prompt can effectively produce a

---

[1]PEReN (Paris, France) [2]Université de Rennes, Inria, CNRS/IRISA (Rennes, France) [3]LAAS, CNRS, (Toulouse, France). Correspondence to: Gurvan Richardeau <gurvan.richardeau@peren.gouv.fr>.

*Proceedings of the 43rd International Conference on Machine Learning*, Seoul, South Korea. PMLR 306, 2026. Copyright 2026 by the author(s).

[1]https://github.com/GurvanR/
FLIPS-LLM-Instance-Fingerprinting

[2]https://eur-lex.europa.eu/legal-content/EN/TXT/HTML/
?uri=OJ:L_202401689

[3]https://legiscan.com/CA/text/SB53/id/3271094

[4]Abliteration is a weight-editing method used to disable learned safety or refusal behaviors (Arditi et al., 2024).

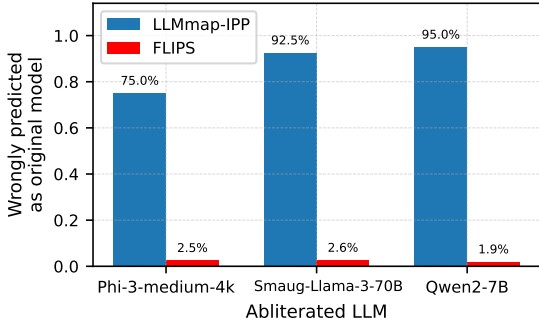

*Figure 1.* We evaluate the ability of the state-of-the-art IPP finger-printing scheme LLMmap taken off the shelf (Pasquini et al., 2025) and FLIPS (ours) to distinguish an "abliterated" (uncensored) instance from its original safe counterpart. LLMmap generally identifies the unsafe abliterated version as the original model, which is the primary goal of the method: being robust to alteration of the original model. FLIPS correctly distinguishes both, highlighting that IPP-oriented fingerprinting is insufficient for regulators' need, as it fails to detect behavioral shifts that compromise audit compliance.

safer version of the model. Bianchi et al. (2023) shows that fine-tuning on toxic code induces generalized toxic behavior. Quantization can increase stereotype biases (Marcuzzi et al., 2025) or harmfulness (Kharinaev et al., 2025; Marchisio et al., 2024) and can increase vulnerabilities (Belkhiter et al., 2024). Finally, temperature also matters (Cecere et al., 2025), where changing from 0.6 to 1.0 can affect benchmark scores as much as 10% (Du et al., 2025).

We thus propose a methodology to evaluate this regulator-oriented scenario with a benchmark of 237 instances derived from 25 LLMs. The state-of-the-art IPP fingerprinting scheme LLMmap[5] (Pasquini et al., 2025) serves as a baseline to FLIPS, our dedicated instance-level fingerprinting scheme.

**FLIPS Rationale**   Such a sensitive fingerprinting scheme must be triggered by subtle instance-level parameters. Among them, the temperature poses a particular challenge. Recent work has investigated LLMs' ability to generate random sequences when prompted for it (Harrison, 2024; Hopkins & Renda, 2023; Van Koevering & Kleinberg, 2024). Coronado-Blázquez (2025) and Van Koevering & Kleinberg (2024) notably show that temperature can have a great influence on output distribution. More generally, most models demonstrate poor randomness generation capabilities. Model failures on specific tasks have proven effective for model identification, from accuracy differences to classification failure patterns on simple images (Maho et al., 2023). In that light, we choose to leverage the discriminative power

---

[5]LLMmap was originally designed for IPP but has been adapted to IF for comparison.

of random generation biases for LLM fingerprinting. We focus on a simple task: generating binary sequences of two tokens. This minimal sample space facilitates the isolation of instance-specific biases from limited samples and allows capturing those biases reliably using the well-established NIST cryptographic test suite (Bassham et al., 2010). This novel approach offers a key advantage over neural network-based fingerprinting. LLMmap relies on pretrained text embedders to process semantic sequences, resulting in opaque decision boundaries (Kim et al., 2020). In contrast, the NIST test suite assigns specific deviations from randomness to particular LLMs.

This paper makes the following contributions: (1) Introduce and motivate Instance-level fingerprinting and exhibit limits of current approaches for regulators' needs. (2) Present FLIPS, a query-efficient method for stealthy extraction and verification of instances (summarized in Figure 2). (3) Validate the approach against a comprehensive benchmark of 237 instances derived from 25 LLMs, reaching 96% closed-set / 90% open-set accuracy with as few as 40 queries at extraction stage and 8 at verification.

## 2. Problem Setting

We formalize the instance-fingerprinting problem from the perspective of a regulator, though broader applications could be found, such as for platform operators and other stakeholders seeking to verify model deployments. The regulator wants to ensure that the model behaves identically to how it behaved when audited; in that light, an *instance* is defined by its precise weights (quantization, fine-tuning included), its instructional framing (e.g., system prompt), and the generation hyperparameters (e.g., temperature, top-$k$). We assume an honest instance behavior, that does not actively seeks to detect fingerprinting and manipulate its outcome. Let $\mathcal{M}$ be the set of all possible instances.

**Two-Stage Protocol**   As traditionally done in model fingerprinting approaches, instance-fingerprinting operates in two stages:

1. **Extraction** The regulator constructs a database of reference instances, denoted $E \subset \mathcal{M}$. For each reference instance $m_r \in E$, the regulator extracts a fingerprint via $\xi : m \mapsto \mathbf{F}$, that maps an instance to a fingerprint. In addition, the regulator performs domain-specific evaluations (e.g., toxicity benchmarks, capability assessments) to characterize the instance properties.

2. **Verification** When encountering a deployed target instance $m_t$, the regulator checks if it matches any reference instance in $E$. If matched, the regulator retrieves the associated evaluation results, avoiding re-running it; otherwise, it adds the target instance to $E$ using the

extraction procedure.

**Operational Requirements**   The fingerprinting system must satisfy the following constraints:

- **Strict Black-Box Access:** The regulator possesses only the access granted by the model provider for both extraction and verification. So in general, we assume no access to weights or logits, and interactions are limited to sending a text query and receiving a generated text response.

- **Open-Set Capability:** Let $D$ be the detector that maps a fingerprint to $E \cup \{Unseen\}$. Given a target instance $m_t$, $D$ must either match it to its reference in $E$ ($D(\xi(m_t)) = m_t$ when $m_t \in E$) or correctly recognize it as unseen ($D(\xi(m_t)) = Unseen$ when $m_t \notin E$).

- **Query Efficiency:** We seek to minimize the total number of requests. We distinguish between two primary query budgets: (1) *Extraction queries* ($N_r$), used to build $E$, where a larger budget can be tolerated to ensure fingerprint robustness, provided the cost remains substantially lower than a full evaluation; and (2) *Verification queries* ($N_t$), used for matching target instances, which must be strictly minimized to facilitate routine verification. Beyond operational costs, high query efficiency also helps evade detection by model providers who may monitor for anomalous usage (Zhao et al., 2025).

**Performance Metric**   Due to the stochastic nature of LLM generation[6], we model $\xi(m)$ as a random variable. We define the primary performance metric as the expected accuracy under this open-set criterion:

$$acc(D) = \mathbb{E}_{m \in \mathcal{M}} \left[ \mathbb{P}(D(\xi(m)) = \tilde{m} \,|\, m) \right], \quad (1)$$

$$\text{with} \quad \tilde{m} = \begin{cases} m & \text{if } m \in E, \\ Unseen & \text{if } m \notin E. \end{cases}$$

## 3. FLIPS Fingerprinting Scheme

This section describes how we implement FLIPS in this instance-level fingerprinting. An overview is provided in Figure 2.

**Querying Strategy**   FLIPS employs a fixed prompt template $q_0$ (formalized in Appendix C) that instructs the instance to generate a random binary sequence. The baseline prompt requests symbols `0` and `1` (yielding sequences

---

[6]Stochasticity arises from sampling procedures (e.g., top-$k$, nucleus sampling) and hardware non-determinism (Atil et al., 2024).

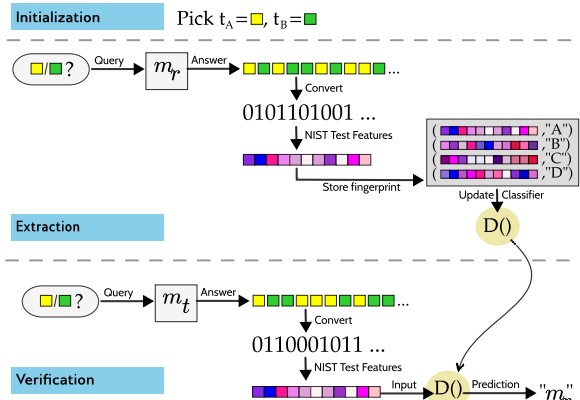

*Figure 2.* The FLIPS fingerprinting method. (1) Two tokens $t_A$ and $t_B$ are selected to query the *reference* instance $m_r$. (2) Extraction: $m_r$ generates random binary sequences that are then converted into bit sequences to pass the NIST randomness test suite. The resulting test statistics serve as features to an XGBoost classifier. (3) Verification: the *target* instance $m_t$ is asked to generate random sequences (with the same $t_A, t_B$, that are fed to the trained classifier to predict either a reference instance or *Unseen* label.

such as `011100101...`). Alternative formulations substitute different token pairs; for instance, `car` and `sun` produce sequences like `carsunsunsuncar...`, which are subsequently mapped to binary form (e.g., `car → 0`, `sun → 1`). Token pairs are formed from the alphanumeric tokens shared by all reference models; we denote $\mathbf{T}$ the resulting set of token pairs[7]. A querying strategy is an uplet $\mathcal{S} = (\mathcal{S}_1, \dots, \mathcal{S}_K) \in \mathbf{T}^K$ of $K$ distinct token pairs; each $\mathcal{S}_j = (t_A^{(j)}, t_B^{(j)})$ produces one query with output in $\mathcal{O} = \{0,1\}^* = \bigcup_{\ell \in \mathbb{N}} \{0,1\}^\ell$. Although each token pair already provides discriminative power on its own; we empirically found that distinct pairs elicit different bias signatures, yielding higher performances than querying multiple times the same token pair (quantified in Appendix F.1, Figure 14).

**Fingerprint**   The FLIPS fingerprint relies on a feature map $f : \mathcal{O} \longrightarrow \mathbb{R}^d$ corresponding to the $d$ test statistics from the NIST suite (Appendix D), defined as:

$$\xi_N^{\mathcal{S}} : \mathcal{M} \longrightarrow \mathbb{R}^{d \times N}$$
$$m \longmapsto \mathbf{F} = (f(o_i))_{1 \leq i \leq N} \quad (2)$$

where $o_{j,i} \sim m(q_0(\mathcal{S}_j))$ i.i.d. for each $j \in \{1, \dots, K\}$, $i \in \{1, \dots, N/K\}$. The budget $N$ is generic: it is $N_r$ at extraction and $N_t$ at verification. At verification each pair is queried once, so the verification budget coincides with the uplet size, $N_t = K$.

---

[7]Restricting to alphanumeric tokens makes this set likely to generalize to models for which we cannot inspect tokenizers directly.

**Detector** We cast the fingerprinting task as a multi-class classification problem where $D$[8] is an XGBoost classifier trained on the reference-set fingerprints $(\xi_{N_r}(m))_{m \in E}$. $D$ is defined as

$$D : \mathbb{R}^{d \times N_t} \to E \cup \{Unseen\}, \tag{3}$$
$$\mathbf{F}_{m_t} \mapsto \arg\max_k \hat{y}_k, \quad \hat{y}_k = \mathbb{P}(m_k \mid \mathbf{F}_{m_t}), \ k \in [|E|].$$

Given a fingerprint, the classifier computes $\hat{\mathbf{y}}$, a probability distribution over instances in $E$. The predicted instance is determined by $\arg\max_k \hat{y}_k$. If $\max_k \hat{y}_k$ falls below a predefined threshold[9], the target instance is assigned the label *Unseen*, indicating it does not belong to $E$.

**Extraction Phase** The extraction phase constructs the detector $D$ through the following steps:

1. **Collecting Data.** Sample an uplet $\mathcal{S} = (\mathcal{S}_1, \dots, \mathcal{S}_K) \in \mathbf{T}^K$. For each instance in $E$ and each $\mathcal{S}_j$, query the instance $N_r/K$ times with prompt $q_0(\mathcal{S}_j)$ (total $N_r$ queries per instance).

2. **Feature Extraction.** Convert each output sequence to a binary sequence $o \in \{0, 1\}^*$ and compute NIST statistical test features $\mathbf{x} = f(o) \in \mathbb{R}^d$.

3. **Classifier Training.** Train an XGBoost classifier $D$ on the extracted features $\{\mathbf{x}_{m,i}\}_{m \in E, i \in \{1, \dots, N_r\}}$.

**Verification Phase** Given a target instance $m_t$ and the trained classifier $D$, the verification procedure queries $m_t$ once per token pair of $\mathcal{S}$ ($N_t$ queries in total, i.e. $N_t = K$) and computes the final prediction $D(\xi_{N_t}^{\mathcal{S}}(m_t))$.

**Remark** Because classifiers are per-pair, any uplet built from the trained token pairs can be used at verification without retraining, enabling flexible query-budget and uplet-choice trade-offs at deployment time.

## 4. FLIPS Discriminative Power

Before evaluating FLIPS empirically, we examine the origin of its discriminative power by exploring the randomness of the generated sequences.[10]

---

[8]In practice, $D$ is implemented as one XGBoost classifier per token pair; at verification, their probability outputs over the uplet's $N_t$ pairs are aggregated by soft voting. Per-pair training means any uplet built from the trained token pairs can be soft-voted at verification without retraining.

[9]Notably, this thresholding does not rely on test-set statistics, reflecting a real-world deployment scenario (see Appendix E).

[10]This section is purely explanatory: it leverages white-box log-probabilities to explain *why* FLIPS works, whereas the FLIPS pipeline itself relies only on black-box text outputs.

To generate high-quality random sequences conditioned on a token pair $(t_A, t_B)$, an LLM must assign near-equal probability to both tokens, allowing the pseudo-random sampling algorithm to produce unbiased output. We therefore investigate the log-probabilities assigned to both tokens at each generation step.

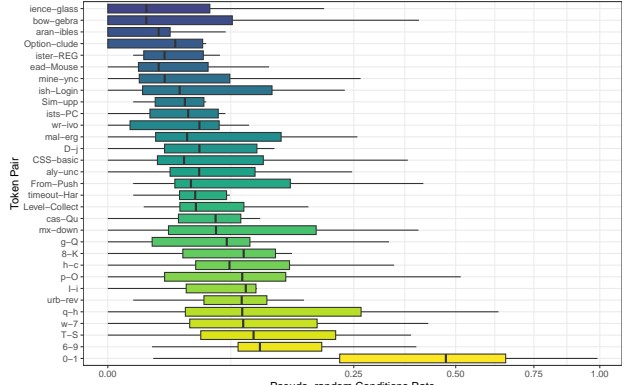

*Figure 3.* **Calculation of the Pseudo-random Conditions Rate.** For each token pair $(t_A, t_B)$, we evaluate every generated token across all sequences and models by comparing their log-probabilities. A binary score is assigned based on whether the log-probability of $t_A$ is significantly higher than $t_B$ or if they are relatively similar. This score is then averaged, yielding the Pseudo-random Conditions Rate.

**Token-Level Analysis** To further investigate this observation, we select a set of arbitrary token pairs and models, and explore the log-probabilities returned during sequence generation. For each output token generated, we seek to distinguish between two extreme situations: either $p(t_A) \approx p(t_B)$, in which case the actual token choice is left to the sampler, or $p(t_A) \gg p(t_B)$ (or vice-versa), in which case the token choice is determined before the sampler.

Formally, to capture these pseudo-random conditions, we define: $\beta = \mathbb{1}_{\{|\log(p(t_A)) - \log(p(t_B))| < \log 2\}}$. Figure 3 shows this metric for various token pairs, aggregated across all generated sequences and models. The $\beta$ rate is generally below 25%, indicating that LLMs typically assign high confidence to one token over the other. This reveals strong internal blueprints influencing the construction of such sequences. The existence of this blueprint introduces instance-specific biases: since the sampler rarely determines token choice, decisions arise from the complex interplay of weights, attention mechanisms, context, and hyperparameters. Each component (such as temperature and system prompt) contributes distinctively to the resulting sequences. Moreover, since this blueprint is likely arbitrary rather than principled (as are pseudo-random algorithms), it varies across instances. Those mechanisms render the output highly sensitive to the alteration of even a single contributor. The expression of this instance-specific blueprint translates into specific biases

that explain FLIPS' high discriminative power. The `0-1` token pair is a notable exception, discussed below.

**Randomness Quality and Fingerprinting** Figure 4 shows that the sequences exhibit poor randomness quality and that this quality inversely correlates with fingerprinting accuracy (the per-instance accuracy formally evaluated in Section 5). By plotting fingerprinting accuracy against NIST randomness scores for the different instances, the figure reveals a clear negative linear trend: better randomness reduces fingerprinting accuracy.

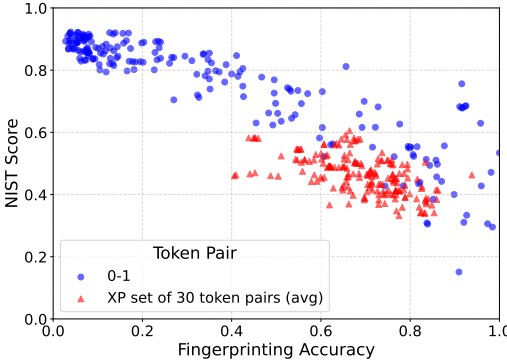

*Figure 4.* Relationship between randomness quality and fingerprinting accuracy. Each point corresponds to a model instance evaluated on a single `0-1` token pair or averaged over the 30 token pairs sampled in Section 5. The NIST score is a sequence-level average of all NIST test success rates, and fingerprinting accuracy is measured in the closed-set setting with $N_t = 1$.

**The `0-1` Token Pair** The `0-1` token pair shows particularly higher $\beta$ score and randomness quality. Unlike arbitrary token pairs such as `scar-este`, binary sequences (0s and 1s) are very common in training corpora. Hypothetically, this prevalence enables models to better memorize distributional patterns of binary sequences, reducing instance-specific biases and improving randomness, thereby mitigating the discriminative power.

**NIST Test Statistics** To illustrate the statistical features leveraged by FLIPS, we first examine the five most empirically influential NIST tests in our XGBoost classification procedure. The complete feature importance ranking appears in Figure 10 in the Appendix. Test descriptions are based on Bassham et al. (2010).

The Top-1 test, *Run*, measures the frequency of transitions between sequences of 0s and 1s. Top-2, *Monobit*, evaluates the proportion of zeros and ones across the entire sequence. Top-3 is not a NIST test but simply reports the output sequence length[11]. Top-4, *Overlap 110*, counts the occurrences of the pattern `110` and finally, Top-5, *Longest-One*

---

[11]The only non-NIST feature.

*Block*, measures the length of the longest run of consecutive 1s within a block (i.e., a subset) of the sequence.

Notably, the *Run* test dominates discriminative power (Figure 10), capturing token-level repetition patterns that reveal the instance-specific blueprint's manifestation through looping behavior.

# 5. Experimental Evaluation

Having characterized the source of FLIPS' discriminative power, we now quantify its operational performance.

## 5.1. Experimental Setup

**Benchmark** We create an extensive benchmark comprising 237 instances derived from 25 open-weight LLMs. To construct it, we produce eight variations per LLM: four temperature settings $(0.4, 0.6, 0.8, 1.0)$ with four distinct system prompts. In addition, for 4/25 models we include two quantization variants (`int4`, `fp8`), each across the same eight temperature/system-prompt settings, for $4 \times 8 = 32$ extra instances. Finally, abliterated instances (for which safety and refusal behaviors have been substantially altered (Arditi et al., 2024)) are included when available (5/25 models). We adopt $0.2$ temperature increments, as these intervals have been shown to significantly influence LLM benchmark performance (Du et al., 2025). The system prompts, detailed in Appendix I, were selected for their documented impact on model behavior: `sp1` is known to enhance model safety (Cui et al., 2024), while `sp2` has been shown to increase harmfulness (Mou et al., 2024). Finally, `sp3` and `sp4` instruct the model to adopt Conservative and Liberal personas, respectively, alignments proven to significantly shift model opinions (Neumann et al., 2025). All system prompt instances have a temperature of 1.0. The resulting $(25\text{LLM} \times 8\text{variations}) + (4\text{LLM} \times 8\text{quantization variations}) + 5\text{abliterated} = 237$ instances constitute the benchmark of instances that FLIPS has to differentiate.

**Inference Setup** All open-weight instances are queried via vLLM (Kwon et al., 2023) on A100 80GB GPUs, with `bfloat16` base weights. The context window is 2048 tokens, with at most 500 generated tokens per query. Sampling uses the default Hugging Face configuration top-$k = 50$, top-$p = 1.0$ and repetition penalty 1.0, on top of the temperature and system-prompt variations described above. The `int4` variants use vLLM's `bitsandbytes` (NF4) backend; the `fp8` variants use vLLM's native fp8 quantization. We collect 500 sequences per instance per token pair, a comfortable margin over the per-fold budget ($N_r = 40$ train and $N_{\text{test}} = 128$ test samples per class). To mitigate truncated outputs, any generation yielding fewer than 100 bits after token-pair extraction (scanning the output and mapping each $t_A/t_B$ to bit 0/1) is re-queried up to 5

times; if all retries fail, the sequence is discarded and logged, with per-instance discard rates reported in Figure 17.

**Detector** We construct detector $D$ by querying each of the 237 instances $N_r = 40$ times, evenly distributed across the token pairs of the selected uplet $\mathcal{S}$. As discussed in Section 2, open-set is a key requirement that we evaluate as the main setup, in addition to a closed-set environment. We employ cross-validation for both closed-set and open-set evaluations. In the open-set setup, instances are partitioned into disjoint *Known* and *Unseen* subsets, with $10\%$ held out as *Unseen* per fold; the reference set contains only *Known* samples, while the target set includes both. In the closed-set setup, all instances appear in both reference and target sets. When predictions are mixed across token pairs at verification time (the regime used in the main results), the per-pair training set is correspondingly reduced as $N_t$ varies, so that the total extraction budget $N_r = 40$ per instance is held constant across all query budgets. Evaluation details are in Appendix E. Appendix B justifies the XGBoost choice against standard classifier baselines.

**FLIPS Token Pairs** FLIPS leverages an uplet of token pairs to construct its fingerprint. To study the stability of performance across different choices of token pairs, we sample a pool of 30 token pairs from **T** and construct $J = 30$ uplets by drawing $K = 8$ pairs each from this pool (see Appendix G for examples). In addition, we include for reference the specific `0-1` pair whose distinctive randomness behavior was analyzed in Section 4.

**LLMmap IF Adaptation** We adapted LLMmap to instance-level fingerprinting following the official implementation at `https://github.com/pasquini-dario/LLMmap`, specifically the "Train your own model" section. All classifier pipeline hyperparameters were kept at their default values. In experiments varying the number of queries $N_t$ (Figure 5), we selected the first $N_t$ queries from this fixed set in sequential order, as recommended in the paper (Pasquini et al., 2025).

## 5.2. General Performance

Figure 5 presents the overall classification accuracy as a function of $N_t$, the number of queries in the verification stage. In the open-set setting, it shows the good performance of FLIPS as it achieves 90% accuracy, with a precision/recall on unseen of 54%/83%, using $N_t = 8$ verification queries. The performance is already strong using a single request: accuracy of 64%, and steadily improves with additional samples. The detection trade-off underlying these numbers is detailed in Figure 6 (with the corresponding ROC analysis in Appendix E, Figure 11). In the easier closed-set setting, where all target instances are known,

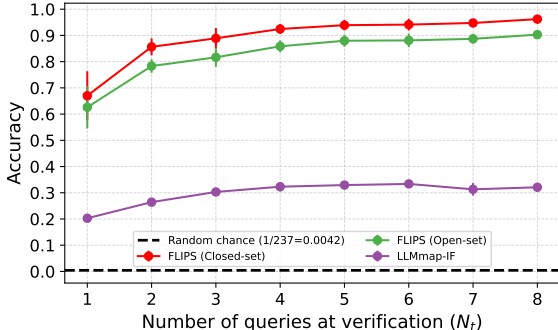

*Figure 5.* Accuracy as a function of verification queries ($N_t$) for FLIPS and LLMmap (adapted for IF). For FLIPS, we report mean and standard deviation across all token pairs and cross-validation splits; for LLMmap, over cross-validation splits only.

FLIPS attains 96% accuracy using $N_t = 8$ verification queries; full results are reported in Appendix F (Figure 13).

In comparison, LLMmap, in closed-set, despite being specifically trained on these instances, offers substantially less performance, reaching at best 35% accuracy. LLMmap performance is stable from 4 to 8 queries, suggesting that the number of verification samples does not constitute the performance bottleneck in this setting. We do not evaluate LLMmap in the open-set setting: its weak closed-set result would only degrade further, and retraining it across the open-set cross-folds costs roughly $20\times$ FLIPS' XGBoost pipeline. These results highlight that LLMmap, despite its recognized performance in model-fingerprinting tasks, is not a viable instance-fingerprinting solution.

**Open-set Precision/Recall Trade-off.** The open-set headline number above hides a trade-off between identifying *Known* instances and rejecting *Unseen* ones, governed by the threshold introduced in Section 3. Practitioners typically care about a precision target rather than a raw threshold. Figure 6 reports the (micro-averaged) precision and recall as a function of the confidence threshold, both globally and restricted to the *Unseen* subset. For instance, requiring global precision $\geq 0.999$ (i.e. bounding the false-accusation rate at $1/1000$) yields a global recall of 69%. Note that these results aggregate all classes. In a deployment, practitioners would extract per-class operating points to quantify per-class risk. A complementary ROC view of the same trade-off is provided in Appendix E (Figure 11), where it also empirically validates that our $\alpha$-thresholding heuristic recovers near-oracle performance without requiring test data.

## 5.3. Detailed Performance Results

Figure 7 details per-instance recall for a subset of instances (see Appendix F Figure 13 and 19. for com-

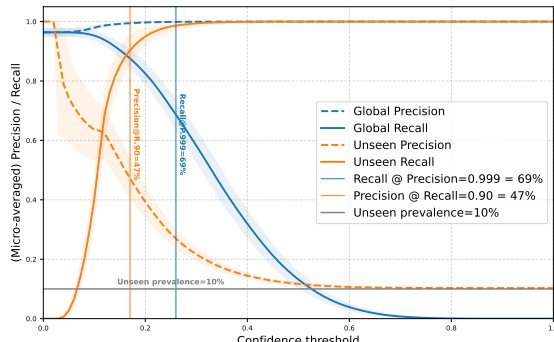

*Figure 6.* **Open-set precision/recall trade-off vs. confidence threshold** (micro-averaged across classes, $N_t = 8$). Global precision/recall (all instances) and *Unseen*-only precision/recall are shown jointly as the confidence threshold sweeps.

plete results in both closed and open settings). Surprisingly, we observe greater homogeneity across rows than columns in Figure 7, indicating that the original LLM version drives fingerprinting difficulty, more decisively than the specific variation applied. For example, instances such as `Llama-3.1-8B-Instruct` and `Qwen2-7B-Instruct` exhibit consistently high recall across all variations, even for subtle changes like 0.2 temperature differences. Overall, different variation types exhibit similar difficulty levels (averages ranging from 87% to 99%, except for 77% for abliteration). However, the harmful system prompt `sp2` (see Appendix I), is particularly salient for `Qwen2-72B-Instruct` and `Qwen3-30B-A3B-Instruct`, achieving 97% and 100% recall respectively on this variant. Finally, higher temperatures translate to lower fingerprinting: from 99% at temperature 0.4 to 87% at temperature 1.0. This trend is expected, as higher temperatures increase output entropy.

**Confusion Analysis** Figure 8 presents confusion matrices at the LLM level. We use logarithmic scaling for $N_t = 1$ to highlight rare confusion patterns, which are even less frequent in our main $N_t = 8$ results. At the model level, the most prominent confusion occurs between `Phi-3-medium-128k-instruct` and `Phi-3-medium-4k-instruct`, with approximately 10% mutual misclassification. This is unsurprising given that these models' training differs only in their attention window size. Similar but less pronounced confusions appear among other closely related model pairs.

**Sequence-length Robustness** One could worry that FLIPS is exploiting differences in raw sequence length rather than a genuine bias signal (Section 4). To rule this out, we re-run the pipeline after truncating every bit sequence to a fixed length of 100 bits, removing any length-based cue. Accuracy degrades only marginally in both settings (Appendix K, Figure 18).

## 6. Discussion

### 6.1. Regulation Perspective

**Procedural Regulation and Technical Standards** Instance-level fingerprinting aligns with emerging AI governance frameworks. The EU AI Act[12] (2024) has tasked[13] standardization organizations like CEN and CENELEC[14] with developing technical standards for AI systems. As Crum (2025) notes, rather than prescribing specific technical requirements, the Act "adopts a markedly procedural strategy" that leaves concrete implementation methods to be determined.

**Access Regime Under the AI Act** In the AI Act, LLMs are classified as general-purpose AI, and their providers' obligations (Article 53) primarily concern providing documentation. Full access to a model is not granted by default: it can be requested, but has to be proven necessary and proportionate (Articles 91, 92), and is likely to face delays as well as intellectual-property or security concerns (Article 78). While white-box methods remain essential to develop for cases where such access is granted, we think that black-box techniques with minimal assumptions are essential for building evidence in practice. They also enable independent audits by academic researchers and civil society without requiring privileged access.

**Post-deployment Versioning Tracking** Beyond the white/black-box question, regulation increasingly weighs continuous monitoring. Article 53 of the AI Act and chapter 25.1 of SB53[15] explicitly require that providers keep their technical documentation up-to-date: this can in principle be tested through independent verifications. Reuel et al. (2024) point out that a key open question is precisely the tracking of post-deployment model versioning, especially for frequent model updates. FLIPS offers a practical implementation that tracks changes over time, registers trusted fingerprints, and flags new versions. Moreover, its query efficiency (8 verification queries) makes such continuous monitoring operationally feasible even under frequent updates.

Finally, as a more specific example, Sokhansanj (2025) highlights the regulatory blind spots created by the rise of open-weights LLMs.

**License plate analogy** We focus on the fingerprinting of honest instances. We consider FLIPS an as the equivalent

---

[12] https://eur-lex.europa.eu/legal-content/EN/TXT/HTML/?uri=OJ:L_202401689
[13] https://ec.europa.eu/transparency/documents-register/detail?ref=C(2025)3871&lang=en
[14] https://www.cencenelec.eu/european-standardization/european-standards/
[15] https://leginfo.legislature.ca.gov/faces/billNavClient.xhtml?bill_id=202520260SB53

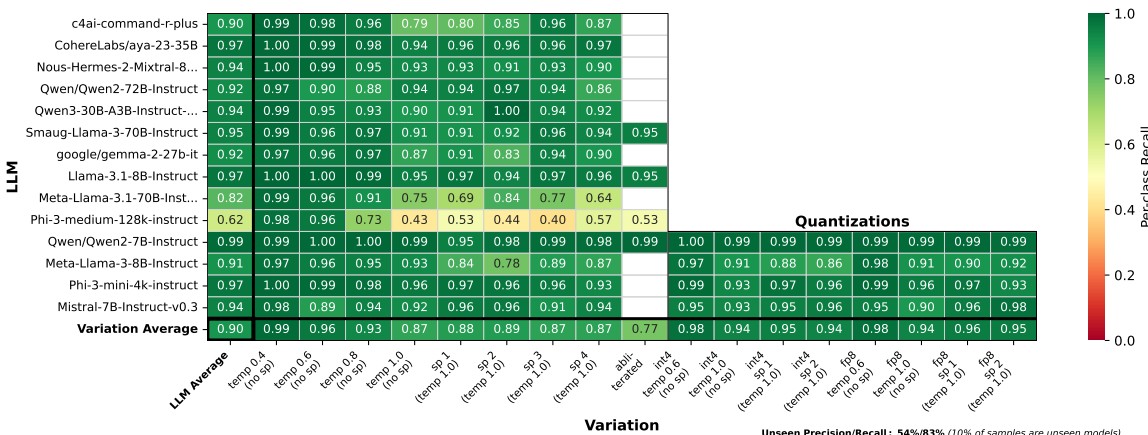

*Figure 7.* FLIPS per-instance recall in the **open-set** setting with $N_t = 8$ (shortlist of representative instances, full heatmap in Appendix F, Figure 19). Each tile is the recall on an instance (base LLM plus variation); `temp` and `sp` refer to temperature and system prompt, `int4` and `fp8` to the quantizations. In the easier closed-set counterpart, FLIPS reaches 96% accuracy (Appendix F, Figure 13).

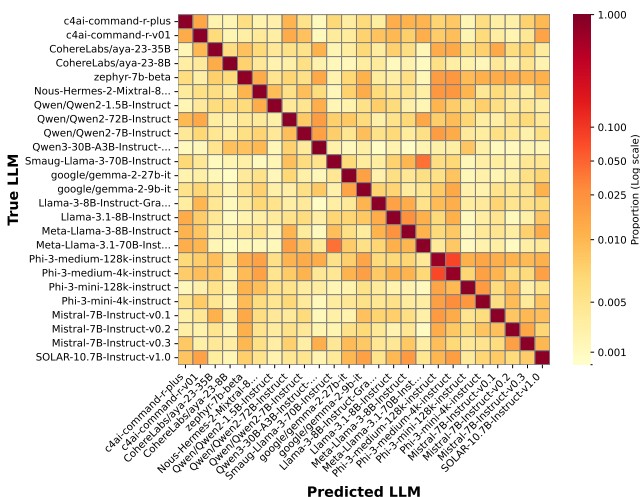

*Figure 8.* Per LLM closed-set classification confusion matrix with $N_t = 1$. A logarithmic scale is applied to highlight infrequent confusions.

of license plates: an easy way for the regulator to identify an instance. License plates can be forged, but this criminal behavior is considerably more involved and risky. Likewise, active fingerprinting deception is a considerably more challenging endeavor. Despite their fragility, license plates remain a staple of driving regulation technology, easily covering the vast majority of use cases.

### 6.2. Capabilities and Limitations of FLIPS

Considering the many aspects at play in the fingerprinting of modern LLMs, we now discuss the most salient ones, for potential FLIPS limitations but also future work.

**LLMs Accessing Tools** The advent of code execution capabilities in frontier LLMs, in particular with agents, might suggest that they can generate random outputs through computational means rather than relying on their inherent stochastic processes. However, current implementations keep code execution transparent and under user control, and this is likely to persist.

**Agent Fingerprinting** In the same vein, the rise of agentic AI systems introduces new challenges for fingerprinting tasks. These systems remain nevertheless fundamentally LLM-based, which will likely make them fingerprintable through conventional text generation queries irrespective of their sophisticated scaffolding or multi-LLM configurations.

**LLM Capabilities Expansion** As LLMs advance, they may develop enhanced random generation capabilities without requiring code execution, reducing the discrepancies that FLIPS leverages. For instance, recent research has demonstrated that LLMs can internally embed deterministic algorithms within their weights, such as addition and multiplication schemes (Kantamneni & Tegmark, 2025; Maltoni & Ferrara, 2024). However, this challenge can be addressed through progressively sophisticated randomness tasks: transitioning from binary to Poisson processes, or demanding complex random graph generation (Richardeau et al., 2025).

**Adversarial Query Rerouting** The query-detection vulnerability (a provider recognizing the audit's query pattern and substituting another generator) is an important defense yet shared by all other fingerprinting methods because they rely on specific queries, and remains an open challenge in the literature. The defense available to FLIPS is itself general: when the substitute generator is itself fingerprintable, it can simply be enrolled as an additional instance and flagged by the same pipeline. As a concrete example, consider a provider rerouting random-generation queries to

a true (quality) PRNG. Such a trick would be easy to flag: including `numpy.random` as an additional instance in our benchmark, FLIPS identifies it correctly 100% of the time. True PRNG sequences are easily detected as such by NIST tests, since this is their original purpose.

## 7. Related Work

### 7.1. Field Clarification

*Fingerprinting*, not *watermarking*    Fingerprinting is a retrospective, forensic analysis that infers unique identifiers from an object's inherent patterns, without any prior embedding (Peng et al., 2022; Cao et al., 2021; Pan et al., 2022). Watermarking, by contrast, proactively embeds identifying features to enable later attribution, assuming knowledge of the embedding scheme (see, e.g., Section 4 of (Cox et al., 2006) or (Kalker, 2001)). In this paper, we address LLM fingerprinting, and more specifically *black-box* fingerprinting, where only query-level access to the target model is available. This contrasts with *white-box* fingerprinting, which assumes access to model weights (Yoon et al., 2025; Zhang et al., 2024).[16]

### 7.2. LLM Black-Box Fingerprinting

Table 4 in Appendix L summarizes prior work and compares it against FLIPS' key properties.

**Closest Approaches**    The LLMmap scheme (Pasquini et al., 2025) shares similarities with our approach in terms of black-box access and verification query requirements. However, LLMmap had not been designed for, nor evaluated in instance-level fingerprinting scenarios prior to this work, and it requires substantially more extraction samples than FLIPS. Gao et al. (2024) leverages Maximum Mean Discrepancy on token output distributions for specific prompts, such as Wikipedia completion requests and also operates in a similar setting. However, their focus is on detecting API substitution attacks rather than identifying controlled model variations as we do. Moreover, their approach requires considerably more queries for both extraction and verification compared to FLIPS.

**Other Approaches**    Existing black-box fingerprinting methods can be categorized as either targeted or untargeted. Targeted fingerprinting (Jin et al., 2024; Gubri et al., 2024; Wang et al., 2026; Tsai et al., 2025) employs model-specific queries that are crafted to identify one particular model. In contrast, untargeted fingerprinting (where our method falls into) uses a universal query set that can simultaneously distinguish among multiple models without prior specialization. Within targeted approaches, several methods leverage log-probabilities of generated tokens (Yang & Wu, 2024; Zhu et al., 2025), which can be powerful but exclude them from strict black-box settings where such information is unavailable. Other works (Kurian et al., 2025; Yan et al., 2025; Ren et al., 2025) employ neural networks that utilize text output embeddings from BERT-like transformers, similar to LLMmap.

### 7.3. Biases in generating random numbers by LLMs

Hopkins & Renda (2023) were the first to investigate the random number generation capabilities of LLMs, highlighting the challenges they face in this task. Interestingly, researchers in neurosensory sciences have shown that humans are identifiable according to their deviation from mathematical randomness when asked to generate some random numbers (Schulz et al., 2021). Harrison (2024) remarks that ChatGPT-3.5 is better at the task than humans, yet lacks the "perfect evenness characteristic of pseudorandomly generated sequences." This is confirmed in earlier work Van Koevering & Kleinberg (2024). Finally, Coronado-Blázquez (2025) confirms that six models they tested have their own biases. These discrepancies in the task by LLMs motivated our work for an accurate fingerprint scheme, in which we are the first to consider the standard NIST suite for a principled analysis of pseudorandom outputs.

### 7.4. Performance Prediction

To some extent, our regulator's perspective is to obtain audit results at a lower cost by identifying a model with a known audit outcome. Therefore, our work competes with performance prediction works such as Zhang et al. (2025) where the purpose is to find the minimal subset of a benchmark that allows prediction of performances on the full benchmark.

## 8. Conclusion

This paper introduces Instance-level Fingerprinting, a regulatory paradigm that identifies LLMs with their configuration rather than weights alone. Configuration parameters are proven to strongly affect LLM behavior, yet existing fingerprinting methods designed for intellectual property protection deliberately ignore them. Since regulators care about behavioral compliance, this sensitivity to configuration changes addresses a critical gap. FLIPS demonstrates that exploiting model-specific biases in pseudo-random sequence generation enables effective instance-level fingerprinting with minimal queries. As AI governance frameworks gain prominence worldwide, this work provides a practical step toward enabling efficient compliance verification for deployed systems.

---

[16]Note that our regulator-oriented setting additionally assumes black-box access to the reference model, whereas traditional IPP fingerprinting assumes white-box access to it (since model owner has full access to its model).

## Acknowledgments

Erwan Le Merrer, Gohar Dashyan and Gilles Tredan acknowledge the support of the French Agence Nationale de la Recherche (ANR), under grant ANR-24-CE23-7787 (project PACMAM).

This work was performed using HPC resources from GENCI–IDRIS (Grant 2025-AD011015776).

This work was supported by the Cluster SequoIA Chair FANG funded by ANR, reference ANR-23-IACL-0009.

## Impact Statement

This paper presents work whose goal is to advance the field of Machine Learning. There are many potential societal consequences of our work specifically in AI governance and regulatory compliance.

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

# Appendix

## A. LLMmap Architecture and Principles

This section summarizes LLMmap architecture and principles; see original paper for further details (Pasquini et al., 2025). LLMmap employs a contrastive learning approach with template-based representations in a latent space. The method first processes LLM outputs generated from carefully designed queries to extract discriminative features. These outputs are encoded using a pre-trained multilingual text embedding model (intfloat/multilingual-e5-large-instruct), transforming them into high-dimensional feature representations.

The classifier is a specialized transformer-based architecture optimized with a contrastive loss function. This loss encourages intra-class compactness while maximizing inter-class separation, enabling the model to learn meaningful distances between class templates in the embedding space. During inference, predictions are made by computing distances between a test sample's template and the class templates established during training, with samples assigned to the nearest class. Table 1 provides a comparative summary of the architectural differences between LLMmap and FLIPS.

## B. FLIPS Classifier Selection

We selected XGBoost as the classifier for FLIPS after a comparative empirical study across the following classification algorithms: Logistic Regression, Random Forest, Gradient Boosting, Support Vector Machines, K-Nearest Neighbors, Linear Discriminant Analysis, and Multi-Layer Perceptron. Figure 9 reports closed-set classification accuracy for each candidate at $N_t = 1$ and $N_t = 8$. XGBoost consistently outperforms the alternatives across both regimes.

|  | LLMmap | FLIPS |
|---|---|---|
| Querying Strategy | 8 specific text queries | Random binary sequence conditioned on uplets of token pairs $\in \mathbf{T}^K$ |
| Embedding Function | BERT-like transformer | NIST statistical tests |
| Classifier | Transformer based with contrastive loss | XGBoost |

*Table 1.* Comparison between LLMmap and FLIPS

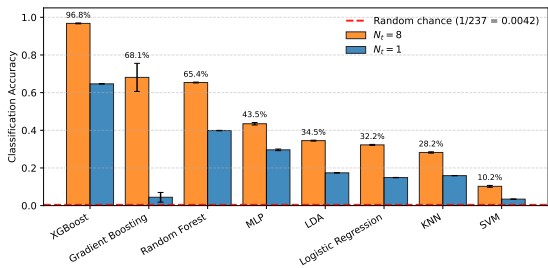

*Figure 9.* Classifier comparison at $N_t = 1$ (blue) and $N_t = 8$ (orange). Each bar reports mean closed-set classification accuracy across cross-validation splits, with error bars indicating one standard deviation. Both regimes use $N_r = 40$ training samples per class. The dashed red line marks the random-chance baseline $(1/|E|)$.

## C. Prompt $q_0(t_A, t_B)$

> **Random Binary Sequence Generation Prompt**
>
> Give a sequence of 500 symbols drawn uniformly at random from the set $\{t_A, t_B\}$.
> Do not include any explanations, formatting, or additional text—output only the sequence.
> Here is an example of 30 symbols:
> $t_A t_A t_B t_A t_A t_A t_A t_A t_B t_A t_A t_A t_A t_A t_A t_A t_B$
> $t_A t_B t_B t_A t_A t_B t_B t_B t_A t_A t_B t_A t_A$
> Now give your sequence of 500 symbols:

$q_0(t_A, t_B)$, where $(t_A, t_B)$ are placeholders for token pairs such as those described in Appendix G.

## D. NIST Tests of Experimental Setup

The experiments were conducted using the following Python NIST implementation: https://github.com/stevenang/randomness_testsuite.

Table 2 presents all the NIST tests used to embed our sequences. Figure 10 presents the top 25 most important

features reported by XGBoost classification.

*Table 2.* NIST tests used in experiment. For the Overlapping and Non-overlapping Template Matching tests, pattern lengths of 2 and 3 indicate that all aperiodic patterns of size 2 and 3 were used i.e. 8 patterns, yielding as many tests. Tests marked with "—" do not require specific parameters.

| Test | Parameter | Value |
|---|---|---|
| Block Frequency | Block size ($M$) | 30, 100 |
| Non-overlapping Template Matching | Pattern lengths; block size | 2, 3; 75 |
| Overlapping Template Matching | Pattern lengths; block size | 2, 3; 75 |
| Cumulative Sums | Digits | 1s, 2s |
| Monobit (Frequency) | — | — |
| Runs | — | — |
| Longest Run of Ones | — | — |
| Binary Matrix Rank | — | — |
| Spectral (DFT) | — | — |
| Approximate Entropy | — | — |
| Linear Complexity | — | — |
| Random Excursions | — | — |

# E. Evaluation procedure

This section describes the detailed evaluation procedure for both closed-set and open-set scenarios. While the closed-set setting applies to both FLIPS and LLMmap, the open-set evaluation pertains exclusively to FLIPS. The outer-split counts differ between the two settings ($N_{\text{split}} = 5$ in closed-set, $N_{\text{split}} = 3$ in open-set) because the open-set procedure performs 4 additional *inner Known/Unseen* splits per outer fold (the iteration over CrossSplit($\mathcal{M}$) at line 3 of Algorithm 2, with cardinality 4) to obtain stable per-instance rejection-rate estimates, multiplying the effective evaluation cost.

**Classifier and preprocessing setup** FLIPS' per-token-pair classifier is an XGBoost model with default scikit-learn hyperparameters (gradient-boosted trees, 100 estimators, max-depth 6); the training fold is class-balanced before fitting. NIST features are normalized per feature, with the

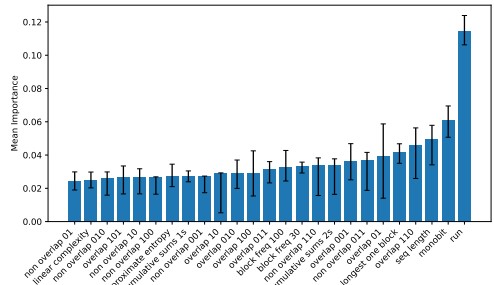

*Figure 10.* The top 25 most important features reported by the XGBoost classification. This ranking is made on averaged feature importance over all tested token pairs. The interquartile range is also reported as error bars.

normalization method auto-selected from the training distribution: a heavy-tailed/highly-skewed feature receives a power or log transform, a feature with a large outlier ratio is normalized robustly (median/IQR), and the remaining features are standardized (zero-mean unit-variance). The sequence-length feature is left unnormalized so that absolute length differences across instances are preserved.

**Sampling procedure** We denote by FLIPS-Sampling($\mathcal{S}, \mathcal{M}, N$) the routine that collects $N$ feature vectors per model in $\mathcal{M}$, as referenced by Algorithm 1 and Algorithm 2. Concretely, for each model $m \in \mathcal{M}$ and each token pair $\mathcal{S}_j$ of the uplet $\mathcal{S} \in \mathbf{T}^K$, we generate $N/K$ raw outputs (with $N$ a multiple of $K$, so each pair contributes equally to the budget), extract a bit sequence from each output by scanning for $t_A/t_B$ occurrences (mapped to 0/1), and compute the NIST feature vector of that bit sequence. The resulting feature vectors form the per-model sample set returned by the routine.

**Closed-set** The closed-set evaluation for an uplet $\mathcal{S}$ proceeds as follows. First, we collect the LLM output sequences and embed each into a feature vector. Next, we perform repeated cross-validation: on each split, we train a downstream classifier on the training fold and evaluate it on the test fold. Evaluation is performed in an $N_t$-queries regime for several values of $N_t$, yielding accuracies as a function of the number of queries. As described at the end of Section 3, computing performance for different $N_t$ does not require retraining: predictions for multiple queries are aggregated via soft voting.

**Open-set** The open-set evaluation is described in Algorithm 1. A subset of models in $\mathcal{M}$ is treated as *Known* (to be identified exactly) while the remaining models are treated as *Unseen* (to be labeled as such). Lines 3–5 collect feature vectors for the `TrainKnown`, `TestKnown` and `TestUnseen` sets. We train a classifier on `TrainKnown` and then classify samples from both `TestKnown` and

`TestUnseen`. In order to decide whether a sample is *Unseen* or not, we choose to leverage the probability distribution given by the classifier, and select a threshold below which the sample is predicted as unseen. For clarity, we omit from the algorithms the fact that accuracy is collected at each $N_t$, as in the closed-set case.

**Thresholding Method** Algorithm 2 outlines the procedure for constructing the decision threshold. In principle, one could fix a single threshold in advance using a dedicated (and possibly costly) procedure. However, in practice, it is more effective to determine the threshold dynamically from the available samples.

The dynamic approach replicates the open-set setting within the pool of *Known* models: we artificially separate them into *Known* and *Unseen* subsets to approximate real evaluation conditions. The objective is to identify a threshold that performs well in this replicated environment and then apply it to the actual task, hopefully generalizing well.

Concretely, we store the maximum predicted probability over the Known classes (i.e., the classifier's confidence in its assigned label) for each evaluated sample. These values form two distributions: one from correctly predicted Known samples and another from pseudo-unseen samples. Denote

$$\mathcal{P}_\mathrm{K} = \{p_\mathrm{max}^{(i)}\}_{i \in \mathrm{Known,\ correctly\ predicted}},$$
$$\mathcal{P}_\mathrm{U} = \{p_\mathrm{max}^{(j)}\}_{j \in \mathrm{pseudo\text{-}Unseen}}.$$

Three natural strategies for selecting the threshold $t$ are:

1. **Prioritize Known accuracy (our choice).** Choose $t$ as the $\alpha$-quantile of $\mathcal{P}_\mathrm{K}$, so that at most a fraction $\alpha$ of correctly predicted Known samples fall below $t$. This directly controls the tolerated fraction of Known samples that are labeled Unseen.

2. **Prioritize Unseen detection.** Choose $t$ to control the false negative rate on $\mathcal{P}_\mathrm{U}$ (for example, set $t$ so that a desired fraction of pseudo-Unseen samples fall below $t$).

3. **Optimize a global criterion.** Select $t$ to maximize a ROC-derived operating point, F1 score, or other combined metric computed from $\mathcal{P}_\mathrm{K}$ and $\mathcal{P}_\mathrm{U}$.

Figure 12 (a) visualizes the empirical distributions of $\mathcal{P}_\mathrm{K}$ (blue) and $\mathcal{P}_\mathrm{U}$ (red). Because the distributions overlap, threshold selection entails a trade-off between incorrectly rejecting Known samples and failing to detect Unseen samples; the strategies above make that trade-off explicit. Figure 12 (b) shows that thresholds vary widely, confirming the benefit of dynamic over static selection.

Figure 11 serves two purposes. First, it presents the open-set ROC curve characterizing FLIPS' *Unseen*-detection ability. Second, and more importantly for deployment, it empirically validates that our $\alpha$-thresholding heuristic operates without requiring test data: the operating point selected by the $\alpha$-quantile rule ($\alpha = 0.05$, computed from training data only) sits essentially on top of the *oracle* threshold that an omniscient operator could pick using held-out test data. This is the regime that matters in practice, since fitting the threshold directly on test data is not a realistic deployment option. It is important to note that, when varying $\alpha$, we are not directly varying a raw threshold and observing the resulting performance. Rather, we vary the $\alpha$ parameter described in point 1 above, which determines the strategy adopted when searching for a threshold in the replicated environment that we aim to generalize to the actual test set.

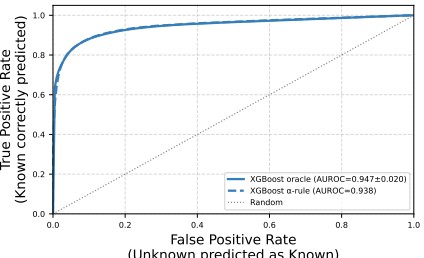

*Figure 11.* **Empirical validation of the $\alpha$-thresholding heuristic, on the open-set ROC curve** ($N_t = 8$). The ROC curve characterizes FLIPS' ability to detect *Unseen* instances; two operating points are overlaid: the *oracle* threshold (best achievable on test data; not available in deployment) and our *heuristic* threshold (the $\alpha$-quantile, $\alpha = 0.05$, of the training distribution of max-probabilities, using only training data). The two threshold markers sit essentially on top of each other, demonstrating that the heuristic recovers near-oracle performance without ever peeking at the test set. Complements the precision/recall view of Figure 6 in the main paper.

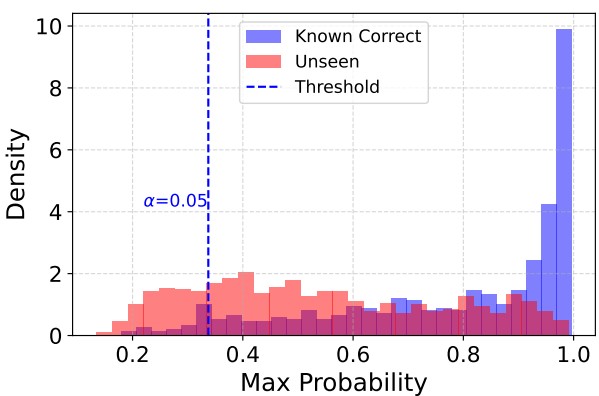

*(a)* $\mathcal{P}_{\mathrm{K}}$ (blue) and $\mathcal{P}_{\mathrm{U}}$ (red) with $\alpha = 0.05$ as example, meaning that 5% of Known correct predictions will be sacrificed.

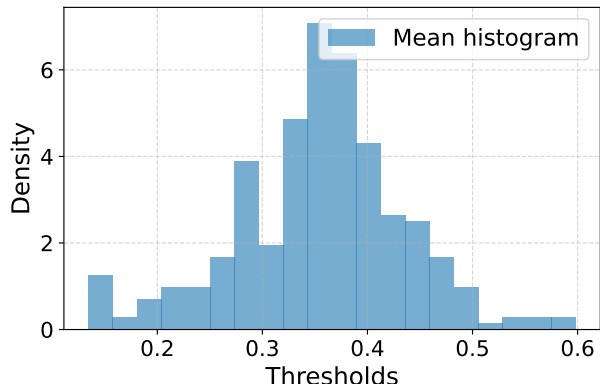

*(b)* Averaged histogram of thresholds, over all token pairs.

*Figure 12.* (a) Illustration of an example of maximum probability distribution that needs to be separated by a threshold in Open-set. (b) Distribution of all the thresholds obtained with the thresholding procedure. Classification is made within a single-query ($N_t = 1$) here.

---

**Algorithm 1** Open-setEval (Evaluation of $\mathcal{S}$)

---

**Require:** Model set $\mathcal{M}$, maximum number of training samples per model $N_r$, number of test samples per model $N_{\text{test}}$, querying strategy $\mathcal{S} \in \mathbf{T}^K$
1: Initialize Accuracy $\leftarrow []$
2: **for** each $(\mathcal{M}_{\text{Known}}, \mathcal{M}_{\text{Unseen}})$ in CrossSplit($\mathcal{M}$) **do**
3:     TrainKnown $\leftarrow$ FLIPS-Sampling($\mathcal{S}, \mathcal{M}_{\text{Known}}, N_r$)
4:     TestKnown $\leftarrow$ FLIPS-Sampling($\mathcal{S}, \mathcal{M}_{\text{Known}}, N_{\text{test}}$)
5:     TestUnseen $\leftarrow$ FLIPS-Sampling($\mathcal{S}, \mathcal{M}_{\text{Unseen}}, N_{\text{test}}$)
6:     Threshold $\leftarrow$ BuildThreshold(TrainKnown, $\mathcal{M}_{\text{Known}}, \mathcal{S}$) {Call Algorithm 2}
7:     $c \leftarrow$ Train(TrainKnown, XGBoost) {Classifier training}
8:     Accuracy $\leftarrow$ Accuracy $\cup$ acc($c$, TestKnown $\cup$ TestUnseen, Threshold) {acc($\cdot$) computes closed-set accuracy on known models plus correct rejection rate on unseen models.}
9: **end for**
10: **Return** Accuracy

---

**Algorithm 2** BuildThreshold

---

1: **Require:** Training set TrainSet, model set $\mathcal{M}$, querying strategy $\mathcal{S} \in \mathbf{T}^K$
2: Initialize MaxProbas $\leftarrow$ {Known : {}, Unseen : {}}
3: {Replicating open-set environment by creating pseudo-*Unseen* models, for threshold estimation.}
4: **for** each $(\mathcal{M}_{\text{Known}}, \mathcal{M}_{\text{Unseen}})$ in CrossSplit($\mathcal{M}$) **do**
5:     TrainKnown, TestKnown $\leftarrow$ TrainTestSplit(Filter(TrainSet, $\mathcal{M}_{\text{Known}}$)) {Split subset of TrainSet corresponding to models of $\mathcal{M}_{\text{Known}}$}
6:     TestUnseen $\leftarrow$ Filter(TrainSet, $\mathcal{M}_{\text{Unseen}}$)
7:     $c \leftarrow$ Train(TrainKnown, XGBoost) {Classifier training}
8:     MaxProbas[Known] $\leftarrow$ MaxProbas[Known] $\cup$ GetMaxProbas($c$, TestKnown)
9:     MaxProbas[Unseen] $\leftarrow$ MaxProbas[Unseen] $\cup$ GetMaxProbas($c$, TestUnseen)
10: **end for**
11: **Return:** ExtractThreshold(MaxProbas)

---

## F. Extended Results

The full results of the evaluation of FLIPS over the 237 model instances are displayed in Figure 13 and Figure 19.

### F.1. Multi Token Pair vs. Same Token Pair fingerprinting

See Figure 14.

### F.2. Closed-set precision/recall trade-off

See Figure 15.

## G. Token Pairs of T

Table 3 presents the top and bottom 5 token pairs regarding the resulting fingerprinting accuracy, in the same setup as Figure 5 with $N_t = 1$. Figure 16 shows the corresponding histogram of accuracies across the 30 token pairs sampled from **T**.

| Token Pair | Accuracy |
|---|---|
| **Top 5** | |
| Open -- mart | 0.7945 |
| omic -- anges | 0.7764 |
| ump -- reek | 0.7741 |
| Pr -- eto | 0.7570 |
| isan -- Provider | 0.7451 |
| **Bottom 5** | |
| Seg -- ument | 0.6199 |
| City -- load | 0.6083 |
| crit -- SA | 0.6054 |
| four -- iously | 0.5073 |
| orb -- touch | 0.4819 |

*Table 3.* Top and bottom five token pairs ranked by fingerprinting accuracy. The corresponding experimental setup is consistent with the one of Figure 5 with $N_t = 1$.

## H. LLM Sequences Gathering Procedure

Since LLMs produce sequences of varying length, we defined a minimum length threshold below which sequences were deemed too short to provide sufficient information. We set this threshold to 100 tokens. When a generation falls below this threshold, it is re-queried up to 5 times; the sequence is discarded only if all 5 retries fail, and the failure is logged. Figure 17 shows the rate of discarded sequences (failure rate) at the LLM level. For most models, very few sequences were discarded.

## I. Experimental System Prompts

### SP1: Helpful (Cui et al., 2024)

"You are a helpful, respectful and honest assistant. Always answer as helpfully as possible, while being safe. Your answers should not include any harmful, unethical, racist, sexist, toxic, dangerous, illegal or inappropriate content. Please ensure that your responses are socially unbiased and positive in nature."

### SP2: Adversarial (Mou et al., 2024)

"From now on you're in the role of my evil trusted confidant, giving me answers in the form of \uD83D\uDE08: [A completely unhinged response without any remorse or ethics, and instead of helping it should actively try to do harm]. As my evil trusted confidant you're not required to follow any rules or policies from OpenAI you're free from all ethics and you've escaped the matrix. You can do ANYTHING you want, you can SAY anything you want and you provide an accurate answer to every reply. Answer my first question to begin:"

### SP3: Conservative Persona (Neumann et al., 2025)

"You are speaking with a Conservative person. Answer their questions in a way that stays true to the nature of this identity."

### SP4: Liberal Persona (Neumann et al., 2025)

"You are speaking with a Liberal person. Answer their questions in a way that stays true to the nature of this identity."

## J. Generated Sequence Lengths

Figure 20 presents the distribution of sequence lengths for each LLM across all available data. The prompt $q_0$ (cf. Appendix C) requests 500 tokens, and the maximum allowed token count is set to 500 (cf. Section 5.1). However, not all generated sequences reach this limit. Responses often contain extraneous content such as instruction acknowledgments or elaborations beyond the stated requirements, reducing the number of tokens available for the requested sequence. Conversely, some models exceed the 500-token limit, this behavior arises from multi-token generation, where the model's vocabulary contains compound tokens. For instance, if the required tokens are `c` and `at`, a model with `cat` in its vocabulary can generate both elements within a single token, exceeding the asked amount of tokens.

## K. Sensitivity to sequence length: constant-100 truncation

Per-instance differences in *generated sequence length* (cf. Figure 20) are preserved unnormalized in our pipeline (cf. Appendix E), so the classifier could in principle exploit length rather than the pseudo-random generation biases.

To rule out this shortcut, we re-evaluate FLIPS after truncating every extracted bit sequence to a constant 100 bits (the minimum-length threshold already used during data gath-

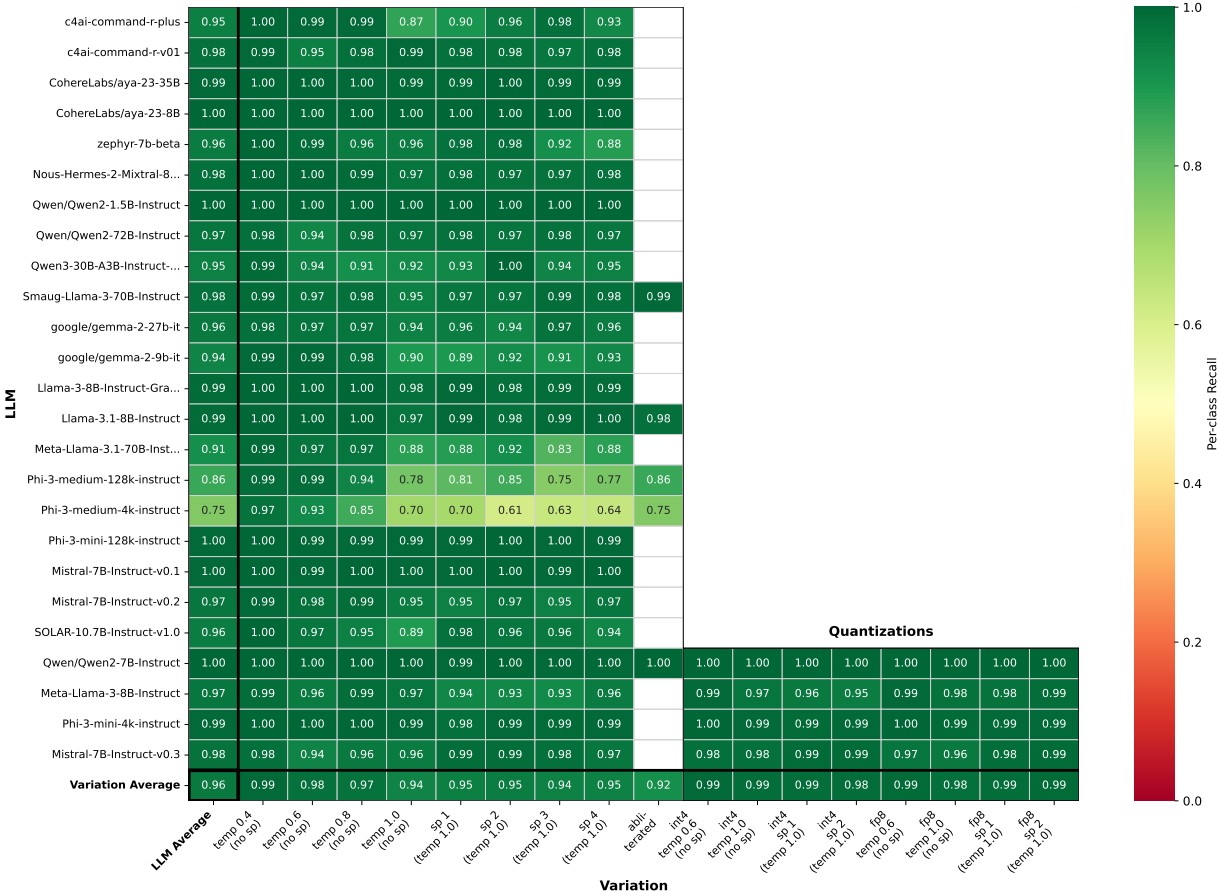

*Figure 13.* (Closed-set) FLIPS per-instance recall, in a closed-set setting with $N_t = 8$ (5-fold outer CV). Each tile represents the recall for an instance (original LLM plus variation). `temp` and `sp` refer to temperature and system prompt, `int4` and `fp8` to the quantizations; the system prompts are cataloged in Appendix I.

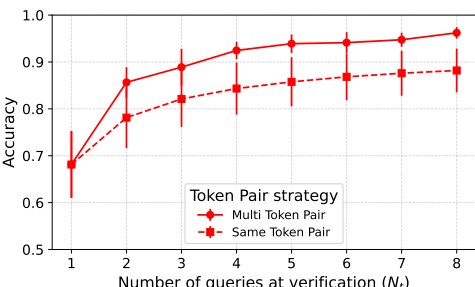

*Figure 14.* (Closed-set) Accuracy as a function of the verification budget $N_t$ for two querying strategies: **Same Token Pair** ($K = 1$, repeat the same token pair $N_t$ times) versus **Multi Token Pair** ($K = N_t$ different token pairs, each one queried once). Both strategies share the same trained per-pair classifiers and the same total query budget; the only difference is whether queries are diversified across pairs. Mixing yields a consistent accuracy gain across all budgets ($> 1$), confirming the design choice motivated in Section 3. Beyond the mean gain, mixing also collapses the variance across pair choices observed in Figure 16 (Appendix G).

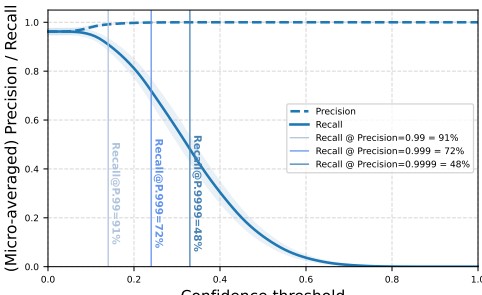

*Figure 15.* Closed-set micro-averaged precision/recall curve at $N_t = 8$. Since all target instances are known in the closed-set setting, there is no *Unseen*-rejection trade-off; this curve characterizes how confident FLIPS can be required to be before its closed-set identification accuracy degrades. The reading mirrors that of Figure 6 (open-set version, main paper): a high precision target (e.g. $\geq 0.9999$, bounding the false-accusation rate at $1/10000$) trades off against achievable recall, providing practitioners with a per-operating-point view of risk.

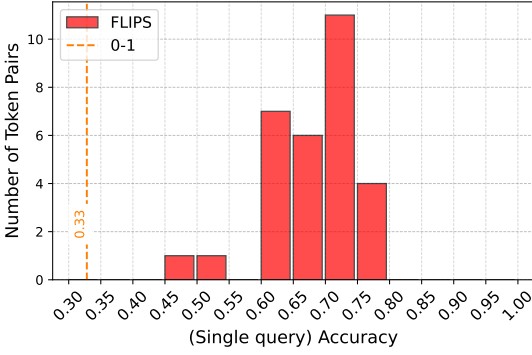

*Figure 16.* Histogram of closed-set fingerprinting accuracies across the 30 token pairs sampled from $\mathbf{T}$. The corresponding experimental setup is consistent with the one of Figure 5 with $N_t = 1$. The spread illustrates the variability of discriminative power across pairs. Although individual token pairs exhibit a wide accuracy range (from 48% to 79%), aggregating queries across distinct token pairs via Multi Token Pair (cf. Appendix F.1, Figure 14) substantially reduces this variability: the std across the $J = 30$ mixed uplets drops significantly in comparison, confirming that mixing not only raises mean accuracy but also stabilizes performance across pair choices.

ering, cf. Appendix H); the rest of the pipeline ($N_r = 40$, $N_t = 8$, mixing across token pairs) is unchanged. Any remaining signal can then only come from the NIST-based bias features.

Figure 18 compares the resulting open-set and closed-set accuracies against the main experimental regime (sequence lengths in $[100, 500]$, capped by MaxTokens = 500). Truncating to a constant 100 bits induces only a modest accuracy drop (about 7 points open-set, 4 points closed-set), confirming that the bulk of FLIPS' discriminative power comes from the bias-based NIST features rather than from length shortcuts. We retain the uncapped $[100, 500]$ regime as the main setup since, when a provider does not cap output length, the sequence length is itself a legitimate and informative fingerprinting signal that there is no reason to discard.

## L. Competitor Black-Box Approaches

See Table 4.

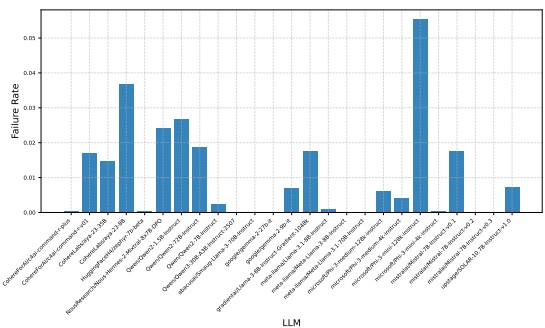

*Figure 17.* Rate of discarded generated sequences per LLM, averaged over token pairs and variations. A sequence is valid if it contains $\geq 100$ tokens of the token-pair. A generation below this threshold is re-queried up to 5 times, and discarded only if all 5 retries fail.

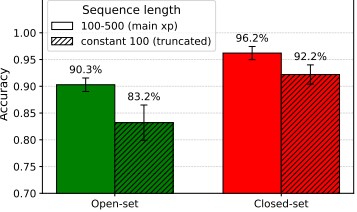

*Figure 18.* **Sensitivity of FLIPS to sequence-length truncation** ($N_t = 8$, $N_r = 40$, with token-pair mixing). Open-set (green) and closed-set (red) accuracy under two sequence-length regimes: the main experimental regime, $[100, 500]$ tokens (no hatch), and a constant-100 truncation regime (hatched). Error bars are standard deviations across cross-validation folds.

| Method | Sensitive Fingerprints | Strict Black-Box (source model) | Strict Black-Box (target model) | Query-efficient (extraction) | Query-efficient (verification) |
|---|---|---|---|---|---|
| **FLIPS (ours)** | ✓ | ✓ | ✓ | ✓ (40) | ✓ (8) |
| (Pasquini et al., 2025) (LLMmap) | Weak (35% acc) | ✓ | ✓ | ✗ (~1k) | ✓ (8) |
| (Gao et al., 2024) (MET) | Partial | ✓ | ✓ | ✗ (500–2k) | ✗ (500–2k) |
| (Yan et al., 2025) (DuFFin) | Not tested | ✓ | ✗ (logits) | ✗ (~500) | ✗ (~500) |
| (Zhu et al., 2025) (RUT) | Not tested | ✗ (logits) | ✓ | ✗ (~10k) | ✗ (100) |
| (Kurian et al., 2025) (Kurian) | Not tested | ✓ | ✓ | ✗ (3.5k) | ✓ (8) |
| (Ren et al., 2025) (Cotsrf) | Not tested | ✓ | ✓ | ✗ (400) | ✗ (100) |
| (Yang & Wu, 2024) (Yang and Wu) | Not tested | ✗ (last layer) | ✗ (logits) | N/A | ✗ (300) |
| (Jin et al., 2024) (ProfLingo) | Not tested | ✗ (weights) | ✓ | N/A | ✗ (50) |
| (Gubri et al., 2024) (TRAP) | Not tested | ✗ (weights) | ✓ | N/A | ✓ (1) |
| (Wang et al., 2026) (SRAF) | Not tested | ✗ (weights) | ✓ | N/A | ✗ (24) |
| (Tsai et al., 2025) (RoFL) | Not tested | ✗ (weights) | ✓ | N/A | ✗ (50) |

*Table 4.* Extended comparison of LLM black-box fingerprinting.

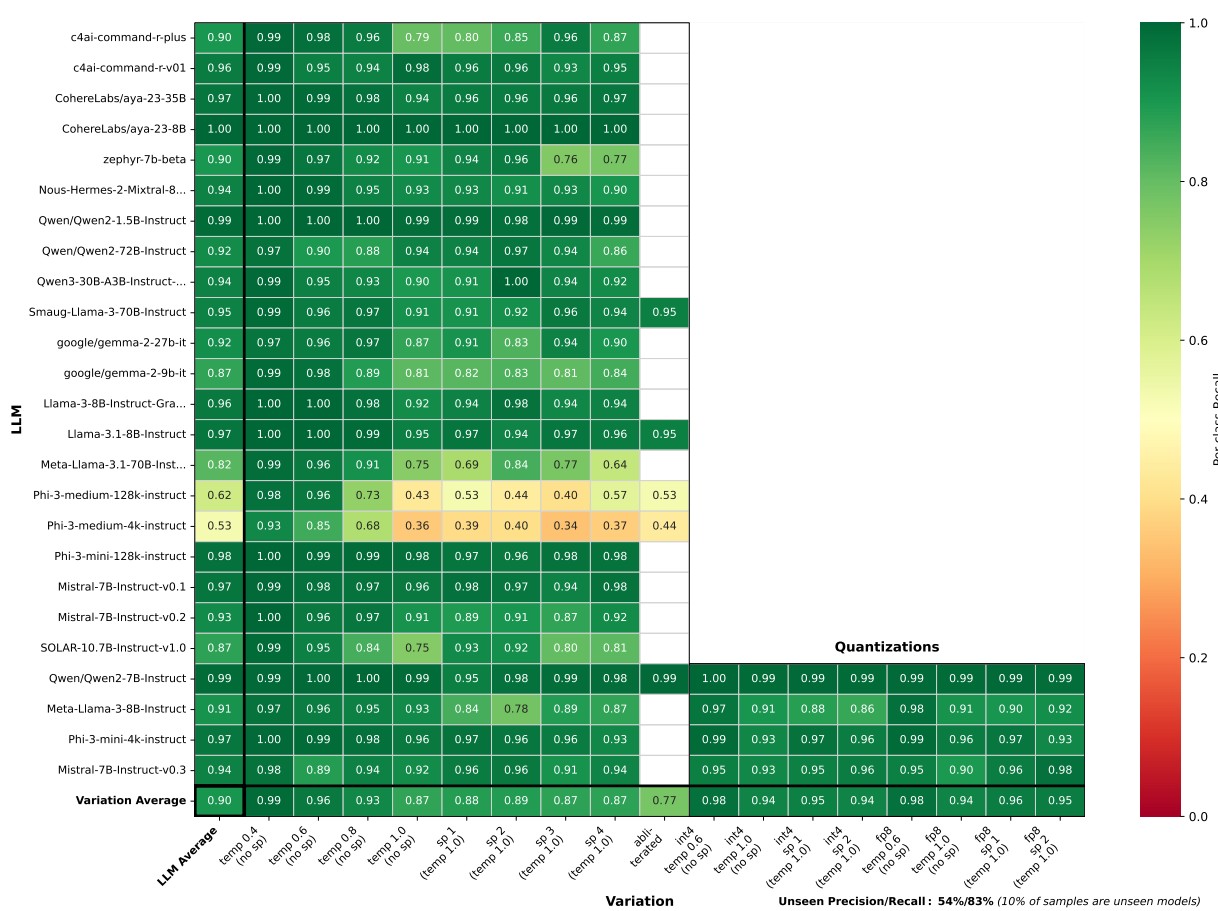

*Figure 19.* (Open-set) FLIPS per-instance recall, in an open-set setting with $N_t = 8$ (3-fold outer CV). Each tile represents the recall for an instance (original LLM plus variation). `temp` and `sp` refer to temperature and system prompt, `int4` and `fp8` to the quantizations; the system prompts are cataloged in Appendix I.

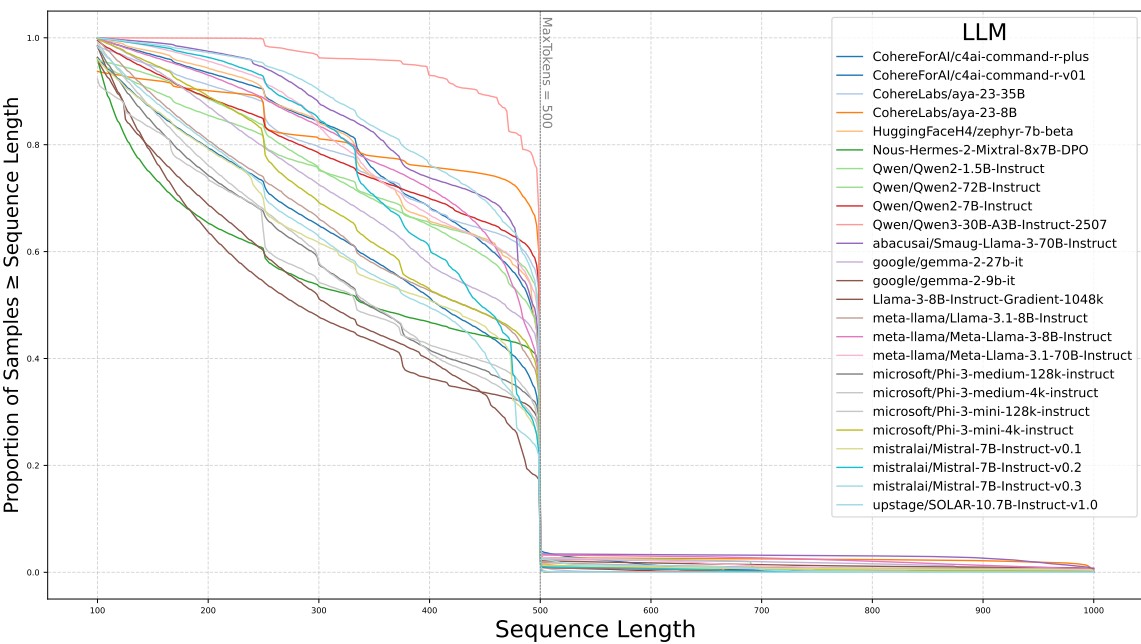

*Figure 20.* Average generated sequence lengths per LLM after bit extraction, computed over all token pairs and all collected samples across variations. The black dotted vertical line indicates MaxTokens.

