# OpenReview forum: "FLIPS: Instance-Fingerprinting for LLMs via Pseudo-random Sequences"
_ICML.cc/2026/Conference — ICML 2026 regular_

### Official Review · Reviewer_1htC · 2026-03-05

**Soundness:** 4
**Presentation:** 4
**Significance:** 3
**Originality:** 4
**Overall Recommendation:** 5
**Confidence:** 4

**Summary:**

This paper introduces instance-level fingerprinting (IF) as a regulator-oriented alternative to intellectual property protection (IPP) fingerprinting, arguing that regulators must identify the actually deployed configuration of an LLM (including system prompt and sampling hyperparameters) rather than only its base weights. The authors propose FLIPS, a black-box, query-efficient method that fingerprints an LLM instance by prompting it to generate pseudo-random binary sequences, extracting statistics from the NIST randomness test suite, and training an XGBoost classifier over these features. On a 205-instance benchmark derived from 25 open-weight LLMs with systematic temperature and system-prompt variations (and several “abliterated” safety-disabled variants), FLIPS achieves up to 90% closed-set accuracy with 8 verification queries, substantially outperforming an adapted LLMmap baseline (35%), and attains 75% accuracy in an open-set setting.

**Compliance With Llm Reviewing Policy:**

Affirmed.

**Final Justification:**

I recommend acceptance as per my initial review; the responses have clarified any remaining concenrs.

**Key Questions For Authors:**

- The “token-level analysis” section uses log-probabilities, which goes beyond strict black-box access (though framed as analysis only). The connection from this white-box probe to deployed black-box settings is a bit unclear: I thought black-box settings were the key focus?
- See limitations below (not too many, but if you could clarify them, that'd be super helpful)
- Finally, can you report open-set metrics beyond accuracy (e.g., AUROC, F1 for unseen detection)?

**Limitations:**

- Baseline coverage is limited to LLMmap; missing empirical comparison to other black-box IPP/IF candidates such as MET (Gao et al., 2024), TRAP (Gubri et al., 2024; albeit targeted), and recent Jacobian-based ZeroPrint. Not a big issue but would be nice to see this.
- Sensitivity to generation settings beyond temperature (e.g., top-p, top-k, repetition penalty), quantization levels, and fine-tuning is mostly discussed but not extensively evaluated. Not a big issue since they evaluate a lot generally. Only so much a single paper can do.

**Strengths And Weaknesses:**

Great work by the authors.
- First, a very nice insight: biases in LLM-generated pseudo-random sequences induced by instance-level parameters, operationalized through NIST test statistics.
- Very clear why randomness-generation biasesa re highly instance-sensitive AND lightweight to elicit.
- Really broad benchmark evaluation (205 instances from 25 LLMs)
- Also super simple to follow and well written
- Demonstrates strong performance with very low query budgets compared to prominent IPP-oriented baselines, showing feasibility for practical oversight workflows.

---

> ### Author Rebuttal · Authors · 2026-03-31
>
> We are very grateful for your enthusiastic review. It is encouraging to hear that you found clear why this random-based method works so well and recognised its low query budgets advantages.
> We first address concerns shared by multiple reviewers in your reply before answering your specific points.
>
> ## Concerns shared by multiple reviewers
>
> Shared concerns ventilated along two groups: 1) the instance fingerprinting scenario, and 2) the evaluation of FLIPS.
>
> ### 1) Regarding the Instance Fingerprinting task
>
> **White/Black box assumption**. In the AI Act, LLMs are classified as general-purpose AI and their obligations (Article 53) are primarily providing documentation. Full access is not granted by default: it can be requested, but has to be proven necessary and proportionate (Articles 91,92), and is likely to face delays and intellectual properties or security concerns (Article 78). While white box methods are essential to develop for such cases, we think black box techniques with minimal assumptions are essential for building evidence. Besides, they can also enable independent audits (academic researchers, civil society).
>
> **Legislation and Deployment Scenario.** Article 53 of the AI act and chapter 25.1 of SB53 explicitly require that providers keep their technical documentation up-to-date: this can be tested with independent verifications. Reuel et al. (2024) point out that a key open-question is the tracking of post-deployment model versioning, especially for frequent model updates: FLIPS offers a practical implementation that tracks changes over time, registers trusted fingerprints, flags new versions. Moreover, FLIPS' query efficiency (8 verification queries) makes continuous monitoring operationally feasible even under frequent updates.
>
> **Adversarial Defense: Provider Rerouting to a True PRNG.** Such a trick with a PRNG would be easy to flag: including numpy.random as an additional instance in our benchmark, FLIPS identifies it correctly 100% of the time. True (quality) **PRNG sequences are easily detected as such by nist tests as it is their original purpose**. In addition, **this query detection vulnerability is shared by all other fingerprinting methods** as they rely on specific queries and remains an open challenge in the literature.
>
> ### 2) Regarding FLIPS evaluation
>
> **Summary of new experimental results:** (1) **token pair mixing**, a simple improvement that boosts performance at no additional query cost; (2) **quantization variants** (FP8 and bitsandbytes-INT4) for 4 models — Mistral-7B-Instruct-v0.3, Phi-3-mini-4k-instruct, Meta-Llama-3-8B-Instruct, Qwen2-7B-Instruct — at 2 temp and 2 system prompts (in total 32 new instances); and (3) **additional metrics** (ROC/Precision/Recall on Unseen/Known and global). Results are with 40 queries at extraction and 8 queries at verification (as in the paper), with the token pair mixing improvement:
>
> | Scenario | Accuracy | Recall on Quantized | AUROC | Unseen Precision/Recall |
> | :--- | :--- | :--- | :--- | :--- |
> | Closed-set | 96.3% | 98.6%  |(Macro-averaged) 0.9996 | N/A |
> | Open-set | 91.5% | 96.9%  | (Unseen vs Known) 0.9530 | 60%/85.3% |
>
> Regarding open-set unseen/known trade-off, as a reminder, a threshold (α) may be calibrated using training data alone to favor recall over precision (i.e. mitigating unseen false miss).
>
> **Token Pair Mixing**. Reviewer 5nBZ questioned whether the performance is sufficient for regulatory purposes. Our point was on the core concept first, but **we can and did improve the method**: rather than drawing all extraction queries from the same token pair, **one can mix different token pairs across queries**. This captures random biases of varied nature at no additional query cost and yields a notable accuracy gain (see table above).
>
> **LLMmap and limited baselines.** To be clear, LLMmap was retrained from scratch on our task, not used off-the-shelf. Almost all other black-box methods are designed for IPP, i.e. built to be robust to the very changes we aim to detect, making comparison structurally unfavorable. MET (Gao et al., 2024) is the only one not framed as IPP, but requires 500–2,000 queries at both extraction and verification, far from our 40/8 setting. This is clarified in the revised paper.
>
> ## Specific to 1htc
>
> **Log-probability analysis and black-box framing.** The β analysis (Section 5) is purely explanatory, a white-box investigation of *why* FLIPS works, not part of the pipeline itself, which requires only raw text outputs. We will clarify this, thank you.
>
> **Comparison to TRAP**. This is a fair point, since TRAP requires white-box access for extraction, we chose to judge it out of scope, which we agree could be discussed.
>
> **ROC on Unseen**. ROC of unseen with its auroc (that you can see on the table) will be added to the revised paper.
>
> We hope the additional metrics and clarifications improved your confidence in the paper. We would be grateful if you considered strengthening your score.

---

> > ### Author Rebuttal · Reviewer_1htC · 2026-04-03
> >
> > Thanks for the authors. I already recommend acceptance, these comments fully satisfy all my issues. Good luck!

---

### Official Review · Reviewer_5nBZ · 2026-03-09

**Soundness:** 2
**Presentation:** 2
**Significance:** 3
**Originality:** 3
**Overall Recommendation:** 3
**Confidence:** 3

**Summary:**

In FLIPS: Instance-Fingerprinting for LLMs via Pseudo-random Sequences, the authors:

- Introduce "instance-level fingerprinting," a problem formulation in which the goal is to distinguish not just base LLMs but specific deployment configurations (weights + system prompt + temperature + quantization), motivated by regulatory compliance scenarios (EU AI Act, California SB 53) where behavioral differences across configurations matter.

- Propose FLIPS, a method that prompts LLMs to generate random binary sequences using arbitrary token pairs, extracts statistical features via the NIST cryptographic randomness test suite, and trains an XGBoost classifier over these features to identify instances.

- Show that LLMs exhibit strong instance-specific biases when generating pseudo-random sequences (the model's internal "blueprint" determines ~75%+ of token choices rather than the sampler), and that these biases are captured by NIST test statistics.

- Evaluate on a benchmark of 205 instances (25 base LLMs × 8 variations across temperature and system prompt, plus 5 abliterated models), achieving 90% closed-set accuracy with 40 extraction queries and 8 verification queries, compared to 35% for the adapted IPP baseline LLMmap.

- Provide an open-set evaluation in which 10% of instances are held out as unseen, achieving 75% overall accuracy with 59% detection of unseen instances.

**Compliance With Llm Reviewing Policy:**

Affirmed.

**Final Justification:**

- Benchmark narrowness was only partially addressed; the dismissal of serving engine, quantization, and hardware variation is unconvincing given that sensitivity to subtle behavioral differences is the paper's entire premise.
- Adversarial robustness was insufficiently addressed: FLIPS relies on a narrow signal band that a knowledgeable adversary could target via calibrated noise, fine-tuning, or prompt-pattern detection.
- The license-plate reframing offered in the rebuttal is reasonable but belongs in the manuscript with an explicit threat model, and currently is not.
- Regulatory framing (threat models, accuracy thresholds, audit workflow, legal admissibility) remains underdeveloped relative to the ambition of the motivating use case.
- The rebuttal did not close the gap between regulatory ambition and evidence base, and reinforces my prior assessment. I maintain my weak reject.

**Key Questions For Authors:**

N/A

**Limitations:**

Yes. Good work. I have already reviewed four papers and not one of them discussed limitations. This is the first to do so, and I have one remaining.

**Strengths And Weaknesses:**

# Strengths

- The core idea is novel and interesting: exploiting LLM-specific biases in pseudo-random sequence generation, featurized via the NIST cryptographic test suite, as a fingerprinting signal. This is a creative repurposing of well-established statistical tools for a new problem.
- The problem formulation (instance-level fingerprinting for regulatory compliance rather than intellectual property protection) is well-motivated and clearly distinguished from prior work. The observation that IPP methods like LLMmap deliberately ignore the configuration-level differences that regulators care about is an important and underappreciated point.
- The 90% accuracy on a 205-way classification problem with only 8 verification queries is a strong empirical result. The method is lightweight (XGBoost on ~25 NIST features), query-efficient, and interpretable relative to neural embedding-based alternatives like LLMmap.
- The β analysis in Section 5, showing that LLMs assign lopsided probabilities ~75%+ of the time during binary sequence generation, provides a clear mechanistic explanation for why the method works. The inverse correlation between randomness quality and fingerprinting accuracy (Figure 8) is a clean and informative finding.
- The comparison table (Table 1) situating FLIPS relative to existing fingerprinting methods across multiple axes (black-box access, query efficiency, sensitivity) is thorough and helpful.

# Weaknesses

## Soundness

- The benchmark is narrower than claimed. The evaluation only varies temperature (4 values), system prompt (4 types), and abliteration (5 models). Quantization, sampling algorithm (top-p, top-k), repetition penalty, serving engine (vLLM, SGLang), speculative decoding, hardware, library versions and fine-tuning are all held fixed. The 205-instance count obscures this: it is 25 models × 8 variations along essentially two axes, plus 5 abliterated models.
- There is no adversarial evaluation. For a method motivated by regulatory compliance, robustness to deliberate evasion is central, not optional. A model provider aware of FLIPS could intercept randomness-generation prompts and route them to a PRNG, or fine-tune to flatten biases on binary generation tasks. The paper does not test or even seriously discuss this threat.
- The paper never establishes what accuracy threshold is actually needed for the regulatory use case. A 10% error rate on a compliance tool (where some errors could mean misidentifying an unsafe instance as a safe one) seems potentially problematic. No analysis of the precision/recall tradeoff at different confidence levels is provided, and asymmetric error costs (false match vs. false miss) are not discussed.
- Only one extraction budget (Nr=40) is tested. No ablation shows degradation at Nr=20 or Nr=10, nor is there analysis of sensitivity to sequence length, NIST test subset, or classifier choice (e.g., k-NN or logistic regression on the same features).

## Presentation

- The paper over-formalizes a simple method. The set-theoretic notation for the fingerprint mapping, output space, and database (Sections 2–3) does no real mathematical work; there are no theorems, bounds, or formal guarantees derived from it. Six pseudocode algorithms in the appendix describe what is essentially "query, extract bits, compute NIST features, train XGBoost, predict with soft voting."
- Figure 3 reports accuracy on a 205-way classification problem without a random-chance baseline. A dashed line at ~0.49% would contextualize both FLIPS (90%) and LLMmap (35%) much more effectively. Without it, readers may interpret the y-axis as if this were a binary task.
- The the β analysis explaining why the method works (Section 5) is buried after the experimental results. Leading with this intuition would make the method click faster for readers.
- Key experimental details (sampling parameters, inference configuration, precision) are relegated to Appendix D rather than being upfront about the scope of what is and is not varied.

## Significance

- The regulatory framing is asserted but underdeveloped. Section 6.1 is three brief sentences about standardization bodies. There is no engagement with what a regulatory deployment would actually require: threat models, legal admissibility, accuracy thresholds, or how FLIPS fits into an audit workflow. The cited legislation (EU AI Act, SB 53) does not prescribe technical requirements, leaving the paper proposing a tool for a use case whose requirements are undefined.
- The discussion of limitations (Section 6.2) raises important concerns — tool use, agentic systems, improving LLM randomness — but addresses each with a short paragraph of speculation rather than experiments. The tool-use concern (LLMs calling a PRNG) appears to trivially defeat the method and is hand-waved.
- The scope of validated variation types (temperature and system prompt) is too narrow to support the general claim of "instance-level fingerprinting." The practical impact depends heavily on whether the method generalizes to quantization, sampling parameters, and infrastructure differences, which are untested.

## Originality

- The connection between pseudo-random generation biases and LLM fingerprinting is novel. Prior work on LLM randomness (Hopkins & Renda, 2023; Coronado-Blázquez, 2025) characterized the biases but did not apply them to identification. The use of the NIST suite as a principled feature extractor for this purpose is a creative and well-motivated choice.
- The instance-level fingerprinting formulation is a useful conceptual contribution, clearly distinguishing the regulatory use case from the IPP setting that dominates prior work.
- However, the methodological novelty is limited in depth: the pipeline is a straightforward composition of known components (prompt engineering -> NIST statistics -> XGBoost), and the paper does not develop theoretical understanding of why specific NIST tests are discriminative for specific types of instance variation.

## Overall

It reads like a solid workshop paper or a short paper that got stretched to full length by adding formalism and appendix algorithms rather than by deepening the actual contribution. The idea is genuinely good but the execution doesn't build enough around it to justify a top venue.

---

> ### Author Rebuttal · Authors · 2026-03-31
>
> **Please first read the reply-to-all in 1htc response.**
>
> To streamline our response, **we first address concerns shared by multiple reviewers in our reply to Reviewer 1htC**. We kindly ask that you refer to that response before reading our point-by-point replies below.
>
> Thank you for your thorough and constructive review. We are glad the instance-level fingerprinting formulation came across as well-motivated and clearly distinguished from prior IPP work, this is really a core contribution. We are also very pleased that the β analysis was recognized as providing a convincing mechanistic explanation for FLIPS's discriminative power.
>
> ## Point-by-point reply
>
> **Serving engine, hardware, library versions.** Our focus is on factors that demonstrably affect what the model outputs and these infrastructure parameters are not known to produce significant differences in LLM behaviors (rather small differences in output distributions). To our knowledge, these parameters have not been the subject of such study.
>
> **Pseudocode verbosity.** We agree. We will retain only Algorithm 3 (open-set evaluation, the non-trivial procedure) and describe the others with more straightforward sentences.
>
> **Random chance baseline in Figure 3.** We will add a dashed line at ~1/N to properly contextualize both FLIPS and LLMmap on a N-classes classification problem.
>
> **Key experimental details upfront.** We will move generation parameters and inference configuration from Appendix D to the main text, by compacting the Related Work section.
>
> **Classifier Choice** We compared the following classifiers: MLP, Random Forest, Logistic Regression, Gradient Boosting, Extra Trees, SVM, KNN, and LDA. XGBoost is the best performer; the second-best (MLP) achieves 5–10% lower accuracy on average and is substantially more resource-intensive. Having the data at hand, we will include the comparison with second best in the appendix and mention other tested methods.
>
> **Precision/Recall trade-off.** Regarding precision/recall tradeoff at different confidence levels, we will plot a figure with (micro-averaged) precision and recall curves as a function of confidence level. Here are values for closed-set setting, indicating for instance that to have 0% False Match rate (i.e., precision = 100%) we need to set the confidence level at 0.4, but decreasing the Recall to only 28% (but 54% only for precision of 99.99%). Thank you for this suggestion, the resulting analysis is very informative and will strengthen the paper.
>
> | Confidence level | Precision | Recall |
> | :--- | :--- | :--- |
> | 0.0 (accuracy) | 0.9695 | 0.9695 |
> | 0.1 | 0.9850 | 0.9545 |
> | 0.2 | 0.9981 | 0.8037 |
> | 0.3 | 0.9999 | 0.5355 |
> | 0.4 | 1.0000 | 0.2759 |
>
> **Sequence length sensitivity**. Please refer to *Constant Sequence Length* paragraph of reply to 53rv.
>
> ## Conclusion
>
> We hope these revisions: new metrics, expanded evaluation (quantization, sequence-length, performance improvement (token-pair mixing)), and the precision/recall analysis, address the gap you identified between the strength of the idea and the depth of its execution. We would be very grateful if you reconsidered your score.

---

> > ### Author Rebuttal · Reviewer_5nBZ · 2026-04-03
> >
> > # To the AC and ICML
> >
> > Some of us have day jobs. It's not acceptable for ICML to demand that reviewers - uncompensated volunteers - respond within 4 workdays.
> >
> > # To the authors
> >
> > I thank the authors for their detailed response.
> >
> > **Concrete fixes (satisfied).** The random-chance baseline, pseudocode reduction, classifier comparison, and moving experimental details into the main text are all straightforward improvements I consider resolved.
> >
> > **Precision/recall analysis (satisfied).** The precision/recall table directly addresses my request and strengthens the paper.
> >
> > **Benchmark narrowness (partially addressed).** I am not persuaded by the dismissal of serving engine and hardware variation. Serving infrastructure can substantially affect LLM outputs (see e.g. the Llama 3 quality variance across vLLM vs Ollama). The entire premise of instance-level fingerprinting is sensitivity to subtle behavioral differences. If FLIPS is robust to infrastructure changes, that is a positive result worth demonstrating; if not, that is a limitation worth acknowledging.
> >
> > **Adversarial evaluation (insufficiently addressed).** The PRNG rerouting experiment is clean but illustrates the fragility rather than resolving it. FLIPS works because LLMs are bad at randomness; the PRNG defense works because NIST detects sequences that are *too* random. The method's security thus depends on a narrow signal band: biased enough to fingerprint, but not so uniform that it looks synthetic. A sophisticated adversary could add calibrated noise atop a PRNG to mimic generic LLM-like biases without preserving instance-specific ones, fine-tune on binary tasks to flatten inter-instance variance, or train a small adapter to detect the prompt pattern. None of these are exotic attacks. A regulatory audit tool should hold even when the adversary fully understands the mechanism; the current defense amounts to detecting the naive countermeasure.
> >
> > **Regulatory framing (partially addressed).** The shared rebuttal cites specific AI Act articles and Reuel et al. (2024) on post-deployment versioning, which is more concrete than Section 6.1 currently offers. I encourage incorporating this into the manuscript. The deeper issues (threat models, legal admissibility, accuracy thresholds, audit workflow integration) remain undiscussed.
> >
> > **Presentation (not addressed).** Leading with the β analysis before experimental results would improve readability.
> >
> > **Updated assessment.** The rebuttal improves the paper on several concrete dimensions. However, my two deepest concerns have been only partially addressed. The adversarial evaluation reveals a structural vulnerability (reliance on a narrow signal band a knowledgeable adversary can target), and the regulatory framing remains underdeveloped in the manuscript. These go to the core of whether the claimed regulatory contribution is sufficiently supported. Overall, a relevant issue considered by this study is timely and the core idea remains creative, but the gap between regulatory ambition and evidence base has not been closed. This study outlines an important domain. I maintain my score.

---

> > > ### Author Response · Authors · 2026-04-08
> > >
> > > Thank you for acknowledging the clarifications.
> > >
> > > Point taken on the presentation, we will put the $\beta$ section right before the experimental one.
> > >
> > > We agree that fingerprinting answers could be altered with calibrated noise or even be rerouted to another LLM entirely. Yet, this defense limitation remains an open problem for all current fingerprinting schemes as they all (to the best of our knowledge) rely on specific queries: LLMmap relies on specific questions, MET on specific Wikipedia completions, etc. If the specific queries are fully disclosed, many defenses stem naturally.
> > >
> > > As an analogy, we perceive Flips as the equivalent of license plates: an easy way for the regulator to identify an instance. License plates can be forged, but this criminal behavior is considerably more involved and risky. Likewise, active fingerprinting deception is a considerably more challenging endeavor. Despite their fragility, license plates remain a staple of driving regulation technology, easily covering the vast majority of use cases.

---

### Official Review · Reviewer_53Rv · 2026-03-11

**Soundness:** 2
**Presentation:** 2
**Significance:** 3
**Originality:** 3
**Overall Recommendation:** 4
**Confidence:** 3

**Summary:**

The paper introduces FLIPS, a fingerprinting technique to distinguish between different model instances. By generating a random sequence of token pairs, evaluating these on the NIST randomness test suite, and using these results as features for a classifier, it is possible for a regulatory body to verify if a specific instance matches any reference instance in a database. The authors test their method on 25 open-weight models with different temperature and system prompts and showcase higher discriminative performance than the baseline LLMmap.

**Compliance With Llm Reviewing Policy:**

Affirmed.

**Final Justification:**

The paper presents a creative methodology, and its query efficiency is a major strength. The results are promising and the authors evaluated the method on a nice suite of models.

The rebuttal also successfully resolved serveral of my concerns. Specifically, the experiments utilizing a constant sequence length were helpful, and my concerns regarding the legal grounding with the EU AI Act were well addressed. I strongly request, however, that this legal context be included in the final version of the paper.

While the novelty is slightly limited, my main concern lies with the experimental scope. Although the suite of models is good, the variations evaluated for each model (e.g., sampling parameters, system prompts) are somewhat narrow. The evaluation really needs to encompass a wider diversity of realistic modifications. I am also not fully convinced of the method's scalability and practicality in real-world deployments with dynamic system prompts and sampling.

**Key Questions For Authors:**

* How does FLIPS perform with combinations of specific configuration settings (different system prompts each with different temperatures)?
* How would FLIPS be practically implemented in real-world deployments, where system prompts and sampling parameters change regularly.
* How does FLIPS compare to other baselines, which were not specifically trained to be robust to different instances?

**Limitations:**

* Impractical Threat model
* Limited evaluation

**Strengths And Weaknesses:**

### Strengths
* Query Efficiency
* Evaluated on many different models
* Creative methodology

### Weaknesses
* **Threat Model:** While technical tools for AI governance will be necessary in the future, this regulatory body threat model is quite limited. The authors heavily motivate their black-box auditing approach with the EU AI Act. However, the specific assumptions how regulations will be practically implemented are somewhat questionable. It is highly unlikely that regulators will rely on pseudo-random token generation to verify model identity. Furthermore, it assumes a regulator with black-box access who must minimize queries to "evade detection by model providers". A legal regulatory body would inherently possess the legal authority to demand exact configurations or white-box access. Moreover, the authors define an "instance" as a combination of weights, system prompt, and hyperparameters. However, in normal model deployments, providers dynamically update system prompt and often adjust sampling temperatures. Using FLIPS, any of these changed would instantly render the model "Unseen". This would be unscalable for real-world deployments.
* **Limited Evaluation:** The experimental setup is quite limited for testing the robustness of FLIPS's discriminative performance. The evaluation utilizes very distinct system prompts and large temperature intervals. It remains unclear how robust FLIPS is when distinguishing between highly similar configurations. Instead of relying on this small dataset, the authors should expand this evaluation or use related datasets like LEAFBench [1].
* **Methodology:**
	* The paper relies solely on LLMmap as a baseline. However, LLMmap was explicitly designed to be robust to system prompts and temperature changes. The authors should include other black-box baselines other than LLMmap. Furthermore, the authors explicitly skip evaluating LLMmap in the open-set setting. If this is too computationally expensive, it should be evaluated on a smaller subset.
	* The authors allow generated sequence lengths to vary between 100 and 500 tokens. Therefore, the classifier may simply be learning the lengths of different instances rather than their pseudo-random generation biases. The evaluation must truncate all sequences to a constant length. Furthermore, the NIST randomness test suites assumes that the sequence length is quite large and not only 100 bits.
	* The paper evaluates 30 token pairs but provides no methodology for how these pairs were selected. Furthermore, taking only 40 samples per instance is statistically very small for measuring randomness, especially at high temperatures.


[1] - Shao et al. SoK: Large Language Model Copyright Auditing via Fingerprinting, https://arxiv.org/pdf/2508.19843

**Presentation**
* Section 4.1: "we produce eight variations per LLM we produce eight variations per LLM:"
* Section 4.3: "see Appendix F 12 for complete results in both closed and open set sttings" --> "see Apendix F Figure 12 for complete results in both closed and open settings"
* Double caption for figure 5 and figure 6.
* Section 7.2: Reference to appendix broken
* Appendix F: "Table 12" --> "Figure 12"
* Section 1: "It consists in extracting a" --> "It consists of extracting a"
* The current structure of the Appendix makes it very hard to read

---

> ### Author Rebuttal · Authors · 2026-03-31
>
> **Please first read the reply-to-all in 1htc response.**
>
> To streamline our response, **we first address concerns shared by multiple reviewers in our reply to Reviewer 1htC**. We kindly ask that you refer to that response before reading our point-by-point replies below.
>
> We appreciate your detailed engagement with our work. We thank you for pushing us to sharpen the regulatory framing and deployment assumptions, addressing these points has led to a more grounded and realistic positioning of FLIPS. We are also glad that the query efficiency, breadth of model coverage, and creativity of the methodology came through.
>
> ## Point-by-point reply
>
> **White/Black box assumption.** For instance, in the AI Act, LLMs are classified as general-purpose AI and their obligations (Article 53) are primarily providing documentation and full access is not granted by default : it can be requested, but has to be proven necessary and proportionate (Articles 91,92), and is likely to face delays and intellectual properties or security concerns (Article 78). While white box methods are essential to develop for such cases, we think black box techniques with minimal assumptions are essential for building evidence. Besides, they can also enable independent audits (academic researchers, civil society).
>
> **Robustness to similar configurations and LEAFBench.** FLIPS is evaluated against configurations known to meaningfully alter model behavior. For instance, the 0.2 temperature step is calibrated on Du et al. (2025) such that it translates into measurable benchmark performance differences. Regarding LEAFBench: it contains 70 instances vs. our 205 (now 237). Ours is thus larger. Moreover, LEAFBench is designed for the IPP setting.
>
> **LLMmap in open-set.** LLMmap substantially underperforms FLIPS in the strictly easier closed-set setting (35% vs. 90%). Open-set performance would be further degraded, and retraining across all open-set cross-validation folds would require ~20× the compute of our full pipeline, which we judged disproportionate given the expected outcome, though we understand the desire for completeness.
>
> **NIST suite and short sequences.** We use NIST statistics as *discriminative features*, not as rigorous randomness certificates. We agree, though, that using NIST statistics as formal randomness certificates on such short sequences would not be rigorous. Thank you for raising this distinction, it is worth making explicit.
>
> **Token Pair Selection**. Please refer to corresponding paragraph in qfdn reply.
>
> **Constant Sequence Length**. Truncating all sequences to a constant 100 bits directly addresses the concern that the classifier might exploit length differences across instances. In the table below are the same results of the table of the reply to all reviewers, but with truncated sequences to 100 bits. Performance remains strong when all sequences are truncated to a constant length (of 100), confirming that **sequence length is not the primary discriminating signal**. That said, we believe both scenarios are worth considering: when a provider does not cap output length, sequence length is a legitimate and informative fingerprinting signal. We will include this new setting in the revised paper.
>
> **The following table is at constant length to 100 bits per sequence:**
> | Scenario   | Accuracy      | Recall on Quantized | AUROC                    | Unseen Precision/Recall |
> |:---------- |:------------- |:---------------- |:------------------------ |:----------------------- |
> | Closed-set | 92.2% (±1.8%) | 98.1% (±1.1%)    | (Macro-averaged) 0.9981  | N/A                     |
> | Open-set   | 83.2% (±3.3%) | 91.9% (±3.3%)    | (Unseen vs Known) 0.9000 | 40%/86.7%               |
>
> **Regulators unlikely to use pseudo-random token generation.** We believe the criterion for adopting an audit tool is whether it meets operational requirements, not the specific mechanism it employs. We elaborate on this in the shared response.
>
> **Presentation flaws.** Finally, we are very grateful for your close attention to the presentation flaws you identified. We will ensure all of them are carefully addressed.
>
> ## Conclusion
>
> We hope that the expanded regulatory grounding, the new quantization and constant-length experiments, and the clarified deployment scenario address the core concerns you raised. We would be very grateful if you reconsidered your score in light of these substantial additions.

---

> > ### Author Rebuttal · Reviewer_53Rv · 2026-04-03
> >
> > Thanks for your rebuttal and the additional experiments. Most of my points were addressed, but I still have some concerns about the scope of the evaluation:
> > * I appreciate the legal grounding using the AI Act, but I am still not fully convinced of the scalability and practicality in real-world deployments with dynamic system prompts or sampling.
> > * While FLIPS evaluates a higher toal number of instances, LEAFBench evaluates a much wider diversity of realistic modifications. While the quantization experiments are interesting, the paper must broaden the evaluation to more variations.
> > * The constant sequence length experiment addresses my concern.
> >
> > Overall, most of my major concerns were adressed. I will be raising my score (2->4)

---

> > > ### Author Response · Authors · 2026-04-08
> > >
> > > We are pleased that the reviewer found our clarifications satisfactory.
> > >
> > > Thank you for acknowledging the legal grounding. Regarding scalability with dynamic configurations: we argue that frequent updates are precisely where FLIPS provides value. When a provider modifies system prompts or sampling parameters, FLIPS flags it as a new instance, which is the desired regulatory behavior, since any undisclosed change should be detected. The key to scalability is that verification requires only 8 queries, making frequent monitoring operationally feasible.

---

### Official Review · Reviewer_qfdn · 2026-03-15

**Soundness:** 3
**Presentation:** 3
**Significance:** 2
**Originality:** 2
**Overall Recommendation:** 4
**Confidence:** 4

**Summary:**

The paper argues that regulators often care about an LLM "instance" (weights + system prompt + sampling parameters + other system-level things) rather than _just the weights_ themselves. This is because changing temperature or prompts can change behaviors like safety, capabilities, refusal boundaries, etc. To audit this properly, the paper proposes FLIPS, which is black-box fingerprinting technique. What FLIPS does conceptually is to ask different LLMs to "flip coins" by printing long strings of two words. Since each LLM is not truly random (and in its own particular way), those mistakes can be used like a fingerprint to tell which LLM version is running.

Concretely, the auditor selects a token pair and prompts each reference instance for many binary random sequences. You then convert the outputs to bits, computes some randomness-test, and trains anmulti-class classifier on the data (XGboost). The paper makes a benchmark of rollouts from open-weight LLMs by varying temperatures and system prompts. The method reaches about ~90% closed-set accuracy and ~75% open-set accuracy with a relatively small number of queries, far outperforming existing baselines.

**Compliance With Llm Reviewing Policy:**

Affirmed.

**Final Justification:**

Overall I feel conflicted on this paper. I think it is overall reasonable science with relative reasonable presentation, but I feel that the significance and originality is a bit low. I am ambivalent and would lean reject if I had to decide.

**Key Questions For Authors:**

How exactly is the common token set built across different tokenizers and languages, and how should one choose a robust token pair?How sensitive are results to poor pairs?

What happens for changes such as quantization? What "breaks" the test?

Can the open-set threshold be chosen without any information from the test distribution, and how stable is it?

**Limitations:**

Yes

**Strengths And Weaknesses:**

Strengths: The core method is nice, make the bot "flip coins" and look at the mistakes (although it bears similarity to other conceptually similar work in testing for idiosyncracies in model generations). I like the motivation of the "instance" vs. "weights" framing i found to be quite timely. I think the method is quite simple and interpretable (randomness stat test + XGBoost) with small query budgets. I could imagine myself implementing this in short order (in fact, I bet Codex could zero-shot a working implementation given the paper's clear explanation). The experiments cover many open models and plausible changes (temperature, prompts, what they call "abliteration"). The log-prob analysis is also helpful.

Concerns: everything is a self-built open-weight benchmark. It lacks study/thinking around more modern productionized agentic systems such as, e.g., Claude Code or Codex. For these modern harnesses, there is a ton of random messages, skills, prompts, etc., dynamically injected into the model's context in the midst of the rollout, and there is a lot to think through regarding how to audit these systems properly. Some of the signal may also just be length/formatting quirks of the inference stack, so it could be brittle or spoofed. There are no strong baselines (less of the authors fault given these don't really exist), e.g., the LLMmap comparison feels unfair for a method built for IP with larger extraction budgets and there are no strong simple baselines (e.g., length-only or monobit/run-only). Robustness to adversarial providers (e.g., ones who also modify temperatures, use different tokenizers, and can easily spot "print 500 random 0/1s" and substitute a true RNG or post-process the output) is left undiscussed and untested.

---

> ### Author Rebuttal · Authors · 2026-03-31
>
> **Please first read the reply-to-all in 1htc response.**
>
> To streamline our response, **we first address concerns shared by multiple reviewers in our reply to Reviewer 1htC**. We kindly ask that you refer to that response before reading our point-by-point replies below.
>
> We sincerely thank you for your thoughtful review. We are glad the instance-vs-weights framing resonated as timely and that the overall method came across as simple and interpretable, these are core goals of our work.
>
> ## Point-by-point reply
>
> **Applicability to agentic systems.** This is certainly an interesting future work, generalization is not straightforward. As a preliminary observation, querying Claude Code with our fingerprinting prompt does produce random sequences with no code execution, suggesting the approach may generalize to at least some agentic configurations. We add this limitation in the appropriate section.
>
> **Open-set threshold selection.** To be fully explicit: the α threshold is chosen entirely from *training data*, with no access to the test distribution. Algorithm 6 replicates the open-set environment by treating a subset of known training models as pseudo-unseen, derives the decision threshold in that replicated setting, and applies it to the actual test set.
>
> **Token Pair Selection**. Token pairs are randomly sampled the set of alphanumeric tokens shared by all reference models. Restricting to alphanumeric tokens makes this set likely to generalize to models for which we cannot inspect tokenizers directly (one can even reduce this set to integers and letters of Latin alphabet to be sure every model shares it). We will state the selection procedure more explicitly in the main text.
>
> **Token Pair Sensitivity**. Regarding differences of performance across token pairs, we provided some of the results in Table 5 of the paper. This table indicates that accuracy on our 30 tested pairs range from 0.79 to 0.96. We will add a histogram of per-pair accuracies in the appendix. Note that optimizing token pair selection may yield significant performance gains but is left for future work to focus on a broader message: any token pair works.
>
> **LLMmap Extraction Query Budget.** Regarding query (extraction) budget, you rightly pointed out that the comparison at equal budget may seem unfair to LLMmap. We also had this point in mind and had ran LLMmap at increasing extraction budgets for completeness:
>
> | Extraction Budget ($N_r$) | LLMmap Accuracy |
> |:---:|:---:|
> | 40 | 35% |
> | 80 | 36% |
> | 120 | 37% |
> | 200 | 38% |
> | 300 | 40% |
>
> Performance slowly increases, remaining well below FLIPS even at 300 queries.
>
> ## Conclusion
>
> We hope these clarifications and new experiments have addressed your remaining concerns. We would be grateful if you considered strengthening your score in light of the added contributions (token pair mixing, quantization, LLMmap scaling analysis).

---

> > ### Author Rebuttal · Reviewer_qfdn · 2026-04-06
> >
> > Thank you!!

---

### Decision · Program_Chairs · 2026-04-30

**Decision:**

Accept (regular)

**Comment:**

The reviewers universally praised the paper's novel problem formulation, which shifts the focus from traditional Intellectual Property Protection (IPP) to "instance-level fingerprinting" (IF) for regulatory compliance. The proposed method, FLIPS, is widely recognized as a creative, query-efficient, and interpretable approach to a timely problem. The paper is well-written, technically sound, and introduces a practical paradigm that will be highly useful to the ICML community focused on AI governance and auditing.

I have carefully read all the reviews, the authors' extensive rebuttals, and the subsequent discussions. The primary points of tension during the review period revolved around the breadth of the evaluation benchmark and the method's robustness against adversarial providers (e.g., the threat of an adversary routing queries to a standard PRNG or fine-tuning the model to output calibrated noise, as astutely raised by Reviewer 5nBZ).

While the concerns regarding adversarial evasion are highly valid, this vulnerability is largely endemic to the current state of query-efficient, black-box fingerprinting. The authors' clarification of their regulatory threat model during the rebuttal, framing the tool as a digital "license plate" that is easy to forge but risky for a compliant provider to spoof, adequately grounds the paper's practical contributions. Furthermore, the authors successfully strengthened the empirical evaluation during the rebuttal phase by adding quantization experiments, addressing concerns about the benchmark's initial narrowness.

Overall, the originality of the problem formulation and the strong initial empirical results provide a valuable foundation for future research. The merits of this work significantly outweigh its limitations, and it is recommended for acceptance.